# Is Heterophily A Real Nightmare For Graph Neural Networks on Performing Node Classification?

## Abstract

Graph Neural Networks (GNNs) extend basic Neural Networks (NNs) by using the graph structures based on the relational inductive bias (homophily assumption). Though GNNs are believed to outperform NNs in real-world tasks, performance advantages of GNNs over graph-agnostic NNs seem not generally satisfactory. Heterophily has been considered as a main cause and numerous works have been put forward to address it. In this paper, we first show that not all cases of heterophily are harmful[1] for GNNs with aggregation operation. Then, we propose new metrics based on a similarity matrix which considers the influence of both graph structure and input features on GNNs. The metrics demonstrate advantages over the commonly used homophily metrics by tests on synthetic graphs. From the metrics and the observations, we find some cases of harmful heterophily can be addressed by diversification operation. With this fact and knowledge of filterbanks, we propose the Adaptive Channel Mixing (ACM) framework to adaptively exploit aggregation, diversification and identity channels in each GNN layer to address harmful heterophily. We validate the ACM-augmented baselines with 10 real-world node classification tasks. They consistently achieve significant performance gain and exceed the state-of-the-art GNNs on most of the tasks without incurring significant computational burden.

## 1 Introduction

Deep Neural Networks (NNs) (LeCun et al., 2015) have revolutionized many machine learning areas, including image recognition (Krizhevsky et al., 2012), speech recognition (Graves et al., 2013) and natural language processing (Bahdanau et al., 2014), *etc.*One major strength is their capacity and effectiveness of learning latent representation from Euclidean data. Recently, the focus has been put on its applications on non-Euclidean data (Bronstein et al., 2016), *e.g.,* relational data or graphs. Combining graph signal processing and convolutional neural networks (LeCun et al., 1998), numerous Graph Neural Networks (GNNs) (Scarselli et al., 2008; Defferrard et al., 2016; Hamilton et al., 2017; Velickovic et al., 2017; Kipf & Welling, 2016; Luan et al., 2019) have been proposed which empirically outperform traditional neural networks on graph-based machine learning tasks, *e.g.,* node classification, graph classification, link prediction and graph generation, *etc.*GNNs are built on the homophily assumption (McPherson et al., 2001), *i.e.,* connected nodes tend to share similar attributes with each other (Hamilton, 2020), which offers additional information besides node features. Such relational inductive bias (Battaglia et al., 2018) is believed to be a key factor leading to GNNs' superior performance over NNs' in many tasks.

Nevertheless, growing evidence shows that GNNs do not always gain advantages over traditional NNs when dealing with relational data. In some cases, even simple Multi-Layer Perceptrons (MLPs) can outperform GNNs by a large margin (Zhu et al., 2020b; Liu et al., 2020; Luan et al., 2020b; Chien et al., 2021). An important reason for the performance degradation is believed to be the heterophily problem, *i.e.,* connected nodes tend to have different labels which makes the homophily assumption fail. Heterophily challenge has received lots of attention recently and there are increasing number of models being put forward to address this problem (Zhu et al., 2020b; Liu et al., 2020; Chien et al., 2021; Zhu et al., 2020a; Yan et al., 2021).

---

[1]In general, harmful heterophily means the heterophilous structure that will make a graph-aware model underperform its corresponding graph-agnostic model.

**Contributions**   In this paper, we first demonstrate that not all heterophilous graphs are harmful for aggregation-based GNNs and the existing metrics of homophily are insufficient to decide whether the aggregation operation will make nodes less distinguishable or not. By constructing a similarity matrix from backpropagation analysis, we derive new homophily metrics to depict how much GNNs are influenced by the graph structure and node features. We show the advantage of our metrics over the existing metrics by comparing the ability of characterizing the performance of two baseline GNNs on synthetic graphs of different levels of homophily. According to the similarity matrix, we observe that diversification operation is able to address some harmful heterophily cases, and based on which we propose Adaptive Channel Mixing (ACM) GNN framework. The experiments on the synthetic datasets, ablation studies and real-world datasets consistently show that the baseline GNNs augmented by ACM framework are able to obtain significant performance boost on node classification tasks on heterophilous graphs.

The rest of this paper is mainly organized as follows: In section 2, we introduce the notation and the background knowledge. In section 3, we conduct node-wise analysis on heterophily, derive new homophily metrics based on a similarity matrix and conduct experiments to show their advantages over the existing homophily metrics. In section 4, we demonstrate the capability of diversification operation on addressing some cases of harmful heterophily and propose the ACM-GNN framework to adaptively utilize the information from different filterbank channels for each node to address heterophily problem. In section 5, we discuss the related works and clarify the differences with our method. In section 6, we provide empirical evaluations on ACM framework, including ablation study and tests on real-world node classification tasks.

## 2   PRELIMINARIES

We will introduce the related notation and background knowledge in this section. We use **bold** fonts for vectors (*e.g., $v$*). Suppose we have an undirected connected graph $\mathcal{G} = (\mathcal{V}, \mathcal{E}, A)$, where $\mathcal{V}$ is the node set with $|\mathcal{V}| = N$; $\mathcal{E}$ is the edge set without self-loop; $A \in \mathbb{R}^{N \times N}$ is the symmetric adjacency matrix with $A_{i,j} = 1$ *iff* $e_{ij} \in \mathcal{E}$, otherwise $A_{i,j} = 0$. We use $D$ to denote the diagonal degree matrix of $\mathcal{G}$, *i.e.,* $D_{i,i} = d_i = \sum_j A_{i,j}$ and use $\mathcal{N}_i$ to denote the neighborhood set of node $i$, *i.e.,* $\mathcal{N}_i = \{j : e_{ij} \in \mathcal{E}\}$. A graph signal is a vector $x \in \mathbb{R}^N$ defined on $\mathcal{V}$, where $x_i$ is defined on the node $i$. We also have a feature matrix $X \in \mathbb{R}^{N \times F}$, whose columns are graph signals and whose $i$-th row $X_{i,:}$ is a feature vector of node $i$. We use $Z \in \mathbb{R}^{N \times C}$ to denote the label encoding matrix, whose $i$-th row $Z_{i,:}$ is the one-hot encoding of the label of node $i$.

### 2.1   GRAPH LAPLACIAN, AFFINITY MATRIX AND THEIR VARIANTS

The (combinatorial) graph Laplacian is defined as $L = D - A$, which is Symmetric Positive Semi-Definite (SPSD) (Chung & Graham, 1997). Its eigendecomposition gives $L = U\Lambda U^T$, where the columns $u_i$ of $U \in \mathbb{R}^{N \times N}$ are orthonormal eigenvectors, namely the *graph Fourier basis*, $\Lambda = \text{diag}(\lambda_1, \ldots, \lambda_N)$ with $\lambda_1 \leq \cdots \leq \lambda_N$, and these eigenvalues are also called *frequencies*. The graph Fourier transform of the graph signal $x$ is defined as $x_\mathcal{F} = U^{-1}x = U^T x = [u_1^T x, \ldots, u_N^T x]^T$, where $u_i^T x$ is the component of $x$ in the direction of $u_i$.

In additional to $L$, some variants are also commonly used, *e.g.,* the symmetric normalized Laplacian $L_{\text{sym}} = D^{-1/2} L D^{-1/2} = I - D^{-1/2} A D^{-1/2}$ and the random walk normalized Laplacian $L_{\text{rw}} = D^{-1} L = I - D^{-1} A$. The graph Laplacian and its variants can be considered as high-pass filters. The affinity (transition) matrices can be derived from the Laplacians, *e.g.,* $A_{\text{rw}} = I - L_{\text{rw}} = D^{-1} A$, $A_{\text{sym}} = I - L_{\text{sym}} = D^{-1/2} A D^{-1/2}$ and are considered to be low-pass (LP) filters (Maehara, 2019). Their eigenvalues satisfy $\lambda_i(A_{\text{rw}}) = \lambda_i(A_{\text{sym}}) = 1 - \lambda_i(L_{\text{sym}}) = 1 - \lambda_i(L_{\text{rw}}) \in (-1, 1]$. Applying the renormalization trick (Kipf & Welling, 2016) to affinity and Laplacian matrices respectively leads to $\hat{A}_{\text{sym}} = \tilde{D}^{-1/2} \tilde{A} \tilde{D}^{-1/2}$ and $\hat{L}_{\text{sym}} = I - \hat{A}_{\text{sym}}$, where $\tilde{A} \equiv A + I$ and $\tilde{D} \equiv D + I$. The renormalized affinity matrix essentially adds a self-loop to each node in the graph, and is widely used in Graph Convolutional Network (GCN) (Kipf & Welling, 2016) as follows,

$$Y = \text{softmax}(\hat{A}_{\text{sym}} \text{ ReLU}(\hat{A}_{\text{sym}} X W_0) \, W_1) \tag{1}$$

where $W_0 \in \mathbb{R}^{F \times F_1}$ and $W_1 \in \mathbb{R}^{F_1 \times O}$ are learnable parameter matrices. GCN can be trained by minimizing the following cross entropy loss

$$\mathcal{L} = -\text{trace}(Z^T \log Y) \tag{2}$$

where $\log(\cdot)$ is a component-wise logarithm operation. The random walk renormalized matrix $\hat{A}_{\mathrm{rw}} = \tilde{D}^{-1}\tilde{A}$, which shares the same eigenvalues as $\hat{A}_{\mathrm{sym}}$, can also be applied in GCN. The corresponding Laplacian is defined as $\hat{L}_{\mathrm{rw}} = I - \hat{A}_{\mathrm{rw}}$. The matrix $\hat{A}_{\mathrm{rw}}$ is essentially a random walk matrix and behaves as a mean aggregator that is applied in spatial-based GNNs (Hamilton et al., 2017; Hamilton, 2020). To bridge the spectral and spatial methods, we use $\hat{A}_{rw}$ in the paper.

## 2.2 METRICS OF HOMOPHILY

The metrics of homophily are defined by considering different relations between node labels and graph structures defined by adjacency matrix. There are three commonly used homophily metrics: edge homophily (Abu-El-Haija et al., 2019; Zhu et al., 2020b), node homophily (Pei et al., 2020), and class homophily (Lim et al., 2021) [2] defined as follows:

$$H_{\mathrm{edge}}(\mathcal{G}) = \frac{\left|\{e_{uv} \mid e_{uv} \in \mathcal{E}, Z_{u,:} = Z_{v,:}\}\right|}{|\mathcal{E}|}, \quad H_{\mathrm{node}}(\mathcal{G}) = \frac{1}{|\mathcal{V}|} \sum_{v \in \mathcal{V}} \frac{\left|\{u \mid u \in \mathcal{N}_v, Z_{u,:} = Z_{v,:}\}\right|}{d_v},$$

$$H_{\mathrm{class}}(\mathcal{G}) = \frac{1}{C-1} \sum_{k=1}^{C} \left[ h_k - \frac{\left|\{v \mid Z_{v,k} = 1\}\right|}{N} \right]_+, \quad h_k = \frac{\sum_{v \in \mathcal{V}} \left|\{u \mid Z_{v,k} = 1, u \in \mathcal{N}_v, Z_{u,:} = Z_{v,:}\}\right|}{\sum_{v \in \{v \mid Z_{v,k} = 1\}} d_v}$$

(3)

where $[a]_+ = \max(a, 0)$; $h_k$ is the class-wise homophily metric (Lim et al., 2021). They are all in the range of $[0, 1]$ and a value close to $1$ corresponds to strong homophily while a value close to $0$ indicates strong heterophily. $H_{\mathrm{edge}}(\mathcal{G})$ measures the proportion of edges that connect two nodes in the same class; $H_{\mathrm{node}}(\mathcal{G})$ evaluates the average proportion of edge-label consistency of all nodes; $H_{\mathrm{class}}(\mathcal{G})$ tries to avoid the sensitivity to imbalanced class, which can cause $H_{\mathrm{edge}}(\mathcal{G})$ misleadingly large. The above definitions are all based on the graph-label consistency and imply that the inconsistency will cause negative effect to the performance of GNNs. With this in mind, we will show a counter example to illustrate the insufficiency of the above metrics and propose new metrics in the following section.

## 3 ANALYSIS OF HETEROPHILY

### 3.1 MOTIVATION AND AGGREGATION HOMOPHILY

Heterophily is believed to be harmful for message-passing based GNNs (Zhu et al., 2020b; Pei et al., 2020; Chien et al., 2021) because intuitively features of nodes in different classes will be falsely mixed and this will lead nodes indistinguishable (Zhu et al., 2020b). Nevertheless, it is not always the case, *e.g.*, the bipartite graph shown in Figure 1 is highly heterophilous according to the existing homophily metrics in equation 3, but after mean aggregation (operated by mean aggregator), the nodes in classes 1 and 2 only exchange colors and are still distinguishable. Authors in (Chien et al.,

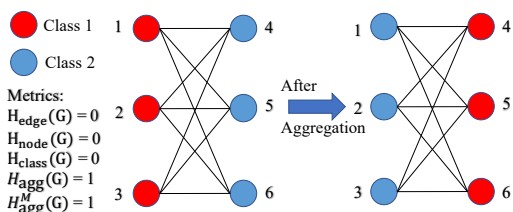

Figure 1: Example of harmless heterophily

2021) also point out the insufficiency of $H_{\mathrm{node}}$ by examples to show that different graph typologies with the same $H_{\mathrm{node}}(\mathcal{G})$ can carry different label information.

To analyze to what extent the graph structure can affect the output of a GNN, we first simplify GCN by removing its nonlinearity as (Wu et al., 2019). Let $\hat{A} \in \mathbb{R}^{N \times N}$ denote a general aggregation operator. Then, equation 1 can be simplified as,

$$Y = \mathrm{softmax}(\hat{A}XW) = \mathrm{softmax}(Y')$$

(4)

---

[2] The authors in (Lim et al., 2021) did not name this homophily metric. We name it *class homophily* based on its definition.

After each gradient decent step $\Delta W = \gamma \frac{d\mathcal{L}}{dW}$, where $\gamma$ is the learning rate, the update of $Y'$ will be (see Appendix C for derivation),

$$\Delta Y' = \hat{A} X \Delta W = \gamma \hat{A} X \frac{d\mathcal{L}}{dW} \propto \hat{A} X \frac{d\mathcal{L}}{dW} = \hat{A} X X^T \hat{A}^T (Z - Y) = S(\hat{A}, X)(Z - Y) \quad (5)$$

where $S(\hat{A}, X) \equiv \hat{A} X (\hat{A} X)^T$ is a post-aggregation node similarity matrix, $Z - Y$ is the prediction error matrix. The update direction of node $i$ is essentially a weighted sum of the prediction error, *i.e.,* $\Delta(Y')_{i,:} = \sum_{j \in \mathcal{V}} \left[ S(\hat{A}, X) \right]_{i,j} (Z - Y)_{j,:}$.

Instead of measuring the graph-label consistency, we study the effect of heterophily by considering the post-aggregation node similarity. To this end, we first define the *aggregation similarity score* as follows.

**Definition 1.** *Aggregation similarity score*

$$S_{agg} \left( S(\hat{A}, X) \right) = \frac{\left| \left\{ v \,|\, \mathrm{Mean}_u(\{S(\hat{A}, X)_{v,u} | Z_{u,:} = Z_{v,:}\}) \geq \mathrm{Mean}_u(\{S(\hat{A}, X)_{v,u} | Z_{u,:} \neq Z_{v,:}\}) \right\} \right|}{|\mathcal{V}|}$$
$$(6)$$

*where* $\mathrm{Mean}_u (\{\cdot\})$ *takes the average over* $u$ *of a given multiset of values or variables.*

$S_{agg}(S(\hat{A}, X))$ measures the proportion of nodes $v \in \mathcal{V}$ that will put relatively larger similarity weights on nodes in the same class than in other classes after aggregation. It is easy to see that $S_{agg}(S(\hat{A}, X)) \in [0, 1]$. But in practice, we observe that in most datasets, we will have $S_{agg}(S(\hat{A}, X)) \geq 0.5$. Based on this observation, we rescale equation 6 to the following modified aggregation similarity for practical usage,

$$S_{agg}^M \left( S(\hat{A}, X) \right) = \left[ 2 S_{agg} \left( S(\hat{A}, X) \right) - 1 \right]_+ \quad (7)$$

In order to measure the consistency between labels and graph structures without considering node features and make a fair comparison with the existing homophily metrics in equation 3, we define the graph ($\mathcal{G}$) aggregation ($\hat{A}$) homophily and its modified version as

$$H_{agg}(\mathcal{G}) = S_{agg} \left( S(\hat{A}, Z) \right), \; H_{agg}^M(\mathcal{G}) = S_{agg}^M \left( S(\hat{A}, Z) \right) \quad (8)$$

In practice, we will only check $H_{agg}(\mathcal{G})$ when $H_{agg}^M(\mathcal{G}) = 0$. As Figure 1 shows, when $\hat{A} = \hat{A}_{\mathrm{rw}}$, $H_{agg}(\mathcal{G}) = H_{agg}^M(\mathcal{G}) = 1$. Thus, this new metric reflects the fact that nodes in classes 1 and 2 are still highly distinguishable after aggregation, while other metrics mentioned before fail to capture such information and misleadingly give value 0. This shows the advantage of $H_{agg}(\mathcal{G})$ and $H_{agg}^M(\mathcal{G})$ by additionally exploiting information from aggregation operator $\hat{A}$ and the similarity matrix.

To comprehensively compare $H_{agg}^M(\mathcal{G})$ with the metrics in equation 3 on how they can reveal the influence of graph structure on GNN performance, we generate synthetic graphs with different homophily levels and evaluate SGC (Wu et al., 2019) and GCN (Kipf & Welling, 2016) on them in the next subsection.

## 3.2 Empirical Evaluation and Comparison on Synthetic Graphs

In this subsection, we conduct experiments on synthetic graphs with different $H_{\mathrm{edge}}^M(\mathcal{G})$ to empirically verify the effectiveness of $H_{agg}^M(\mathcal{G})$ and compare it with the existing metrics.

**Data Generation & Experimental Setup** We first generate 280 graphs in total with 28 edge homophily levels varied from 0.005 to 0.95, each corresponding to 10 graphs. For every generated graph, we have 5 classes with 400 nodes in each class. For nodes in each class, we randomly generate 800 intra-class edges and $[\frac{800}{H_{\mathrm{edge}}(\mathcal{G})} - 800]$ inter-class edges. The features of nodes in each class are sampled from node features in the corresponding class of 6 base datasets (*Cora, CiteSeer, PubMed, Chameleon, Squirrel, Film*). Nodes are randomly splitted into 60%/20%/20% for train/validation/test. We train 1-hop SGC (*sgc-1*) (Wu et al., 2019) and GCN (Kipf & Welling, 2016) on the synthetic graphs [3]. For each value of $H_{\mathrm{edge}}(\mathcal{G})$, we take the average test accuracy and standard deviation of

---

[3]See Appendix A.1 for hyperparameter searching range and Appendix B for more detailed description of the data generation process

runs over 10 generated graphs. For each generated graph, we also calculate its $H_{\text{node}}(\mathcal{G}), H_{\text{class}}(\mathcal{G})$ and $H_{\text{agg}}^M(\mathcal{G})$. Model performance with respect to different homophily values are shown in Figure 2.

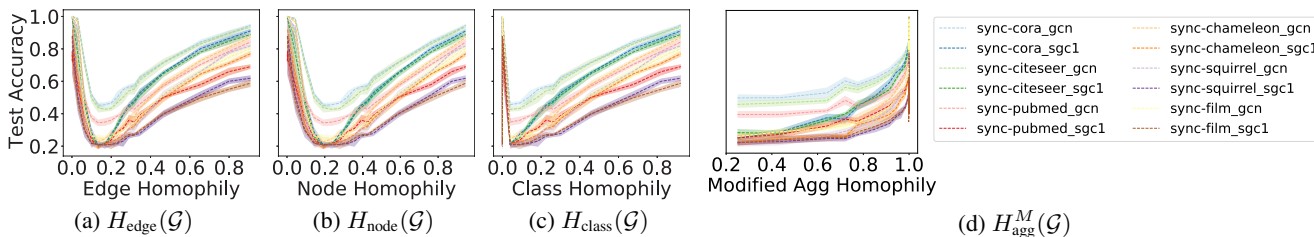

Figure 2: Comparison of baseline performance under different homophily metrics.

**Comparison of Homophily Metrics**   The performance of SGC-1 and GCN are expected to be monotonically increasing with a proper and informative homophily metric. However, Figure 2(a)(b)(c) show that the performance curves under $H_{\text{edge}}(\mathcal{G}), H_{\text{node}}(\mathcal{G})$ and $H_{\text{class}}(\mathcal{G})$ are $U$-shaped [4], while Figure 2(d) reveals a nearly monotonic curve with a little perturbation around 1. This indicates that $H_{\text{agg}}^M(\mathcal{G})$ can describe how the graph structure affects the performance of SGC-1 and GCN more appropriately and adequately than the existing metrics. (See more discussion on aggregation homophily and theoretical results for regular graphs in Appendix B.)

## 4   ADAPTIVE CHANNEL MIXING (ACM) FRAMEWORK

High-frequency graph signal, which can be extracted by high-pass filter, is empirically shown to be useful for addressing heterophily problem (Luan et al., 2020b; Chien et al., 2021; Bo et al., 2021). In this section, based on the similarity matrix proposed in equation 5, we will figure out how diversification operation *i.e.,* highpass (HP) filter, is potentially capable to address

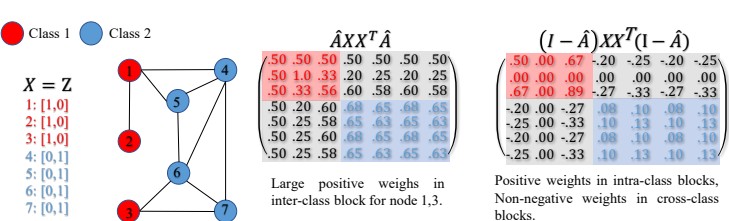

Figure 3: Example of how diversification operation addresses harmful heterophily

some cases of harmful heterophily. From the node-wise analysis, we argue that aggregation (LP filter) and diversification (HP filter) should be combined together as a filterbank (Ekambaram, 2014) for feature extraction and different nodes may have different needs for the information processed by different filters. Based on the above argument, we generalize filterbank method and propose Adaptive Channel Mixing framework in subsection 4.2 to address heterophily challenge.

### 4.1   HOW DIVERSIFICATION OPERATION HELPS WITH HARMFUL HETEROPHILY

We first consider the example shown in Figure 3. From $S(\hat{A}, X)$, we can see that nodes 1,3 assign relatively large positive weights to nodes in class 2 after aggregation, which will make node 1,3 hard to be distinguished from nodes in class 2. Despite the fact, we can still distinguish nodes 1,3 and 4,5,6,7 by considering their neighborhood differences: nodes 1,3 are different from most of their neighbors while nodes 4,5,6,7 are similar to most of their neighbors. This indicates, in some cases, although some nodes become similar after aggregation, they are still distinguishable via their surrounding dissimilarities. This leads us to use *diversification operation*, *i.e.,* HP filter $I - \hat{A}$ (Ekambaram, 2014) to extract the information of neighborhood differences and address harmful heterophily. As $S(I - \hat{A}, X)$ in Figure 3 shows, nodes 1,3 will assign negative weights to nodes

---

[4]A similar J-shaped curve for $H_{\text{edge}}(\mathcal{G})$ is found in (Zhu et al., 2020b), though using different data generation processes. The authors do not mention the insufficiency of edge homophily.

4,5,6,7 after diversification operation, *i.e.,* nodes 1,3 treat nodes 4,5,6,7 as negative samples and will move away from them during backpropagation. This example reveals that there exist some cases that diversification operation is helpful while aggregation operation is not. Based on this observation, we first define diversification distinguishability of a node and graph diversification distinguishability value as follows to measure the proportion of nodes that diversification operation is potentially helpful for.

**Definition 2.** *Diversification Distinguishability (DD) based on* $S(I - \hat{A}, X)$.

*Given* $S(I - \hat{A}, X)$, *a node* $v$ *is diversification distinguishable if the following two conditions are satisfied at the same time,*

$$
\begin{aligned}
&\textit{1. } \mathrm{Mean}_u\left(\{S(I - \hat{A}, X)_{v,u} | u \in \mathcal{V} \wedge Z_{u,:} = Z_{v,:}\}\right) \geq 0; \\
&\textit{2. } \mathrm{Mean}_u\left(\{S(I - \hat{A}, X)_{v,u} | u \in \mathcal{V} \wedge Z_{u,:} \neq Z_{v,:}\}\right) \leq 0
\end{aligned}
\tag{9}
$$

*Then, graph diversification distinguishability value is defined as*

$$
\mathrm{DD}_{\hat{A},X}(\mathcal{G}) = \frac{1}{|\mathcal{V}|}\left|\{v | v \in \mathcal{V} \wedge v \textit{ is diversification distinguishable}\}\right|
\tag{10}
$$

We can see that $\mathrm{DD}_{\hat{A},X}(\mathcal{G}) \in [0, 1]$. Based on definition 2, the effectiveness of diversification operation on addressing heterophily can be proved for binary classification problems under certain conditions, leading us to:

**Theorem 1.** (See Appendix E for proof). For a binary classification problem, *i.e.,* $C = 2$, suppose $X = Z, \hat{A} = \hat{A}_{\mathrm{rw}}$. Then for any $I - \hat{A}_{\mathrm{rw}}$, all nodes are diversification distinguishable and $\mathrm{DD}_{\hat{A},Z}(\mathcal{G}) = 1$.

Theorem 1 theoretically demonstrates the importance of diversification operation on handling heterophily. Combined with aggregation operation, we can get a filterbank which uses both LP and HP filters to distinctively extract the low- and high-frequency information from graph signals. We will introduce filterbank in the next subsection.

### 4.2 FILTERBANK AND ADAPTIVE CHANNEL MIXING (ACM) FRAMEWORK

**Filterbank** For the graph signal $x$ defined on $\mathcal{G}$, a 2-channel linear (analysis) filterbank (Ekambaram, 2014) [5] includes a pair of low-pass (LP) and high-pass (HP) filters $H_{\mathrm{LP}}, H_{\mathrm{HP}}$, where $H_{\mathrm{LP}}$ and $H_{\mathrm{HP}}$ retain the low-frequency and high-frequency content of $x$, respectively.

Most existing GNNs are under uni-channel filtering architecture (Kipf & Welling, 2016; Velickovic et al., 2017; Hamilton et al., 2017) with either $H_{\mathrm{LP}}$ or $H_{\mathrm{HP}}$ channel that only partially preserves the input information. Unlike the uni-channel architecture, filterbanks with $H_{\mathrm{LP}} + H_{\mathrm{HP}} = I$ will not lose any information of the input signal, *i.e.,* perfect reconstruction property (Ekambaram, 2014). Generally, the Laplacian matrices ($L_{\mathrm{sym}}, L_{\mathrm{rw}}, \hat{L}_{\mathrm{sym}}, \hat{L}_{\mathrm{rw}}$) can be regarded as HP filters (Ekambaram, 2014) and affinity matrices ($A_{\mathrm{sym}}, A_{\mathrm{rw}}, \hat{A}_{\mathrm{sym}}, \hat{A}_{\mathrm{rw}}$) can be treated as LP filters (Maehara, 2019; Hamilton, 2020). Moreover, it can be interpreted that MLPs uses the identity filterbank with $H_{\mathrm{LP}} = I$ and $H_{\mathrm{HP}} = 0$ that satisfy $H_{\mathrm{LP}} + H_{\mathrm{HP}} = I + 0 = I$.

From Figure 3, we also observe that different nodes may have different needs for the information from different channels, *e.g.,* nodes 1,3 demand information from HP channel while node 2 only needs information from LP channel. To adaptively leverage the LP, HP and identity channels in GNNs, we propose the Adaptive Channel Mixing (ACM) architecture in the following part.

**Adaptive Channel Mixing (ACM) Framework** ACM framework can be applied to lots of baseline GNNs and in this part, we use GCN as an example to introduce ACM framework in matrix form. We use $H_{\mathrm{LP}}$ and $H_{\mathrm{HP}}$ to represent general LP and HP filters. The ACM framework includes 3 steps as

---

[5]In graph signal processing, an additional synthesis filter (Ekambaram, 2014) is required to form the 2-channel filterbank. But synthesis filter is not needed in our framework, so we do not introduce it in our paper.

follows,

**Step 1. Feature Extraction for Each Channel:**

Option 1: $H_L^l = \text{ReLU}\left(H_{\text{LP}}H^{l-1}W_L^{l-1}\right)$, $H_H^l = \text{ReLU}\left(H_{\text{HP}}H^{l-1}W_H^{l-1}\right)$, $H_I^l = \text{ReLU}\left(IH^{l-1}W_I^{l-1}\right)$;

Option 2: $H_L^l = H_{\text{LP}}\text{ReLU}\left(H^{l-1}W_L^{l-1}\right)$, $H_H^l = H_{\text{HP}}\text{ReLU}\left(H^{l-1}W_H^{l-1}\right)$, $H_I^l = I\,\text{ReLU}\left(H^{l-1}W_I^{l-1}\right)$;

$H_0 = X \in \mathbb{R}^{N \times F_0}$, $W_L^{l-1}, W_H^{l-1}, W_I^{l-1} \in \mathbb{R}^{F_{l-1} \times F_l}$;

**Step 2. Row-wise Feature-based Weight Learning**

$\tilde{\alpha}_L^l = \sigma\left(H_L^l \tilde{W}_L^l\right)$, $\tilde{\alpha}_H^l = \sigma\left(H_H^l \tilde{W}_H^l\right)$, $\tilde{\alpha}_I^l = \sigma\left(H_I^l \tilde{W}_I^l\right)$, $\tilde{W}_L^{l-1}, \tilde{W}_H^{l-1}, \tilde{W}_I^{l-1} \in \mathbb{R}^{F_l \times 1}$

$\left[\alpha_L^l, \alpha_H^l, \alpha_I^l\right] = \text{Softmax}\left(\left(\left[\tilde{\alpha}_L^l, \tilde{\alpha}_H^l, \tilde{\alpha}_I^l\right]/T\right)W_{\text{Mix}}^l\right) \in \mathbb{R}^{N \times 3}$, $T \in \mathbb{R}$ temperature, $W_{\text{Mix}}^l \in \mathbb{R}^{3 \times 3}$;

**Step 3. Node-wise Adaptive Channel Mixing:**

$$H^l = \text{diag}(\alpha_L^l)H_L^l + \text{diag}(\alpha_H^l)H_H^l + \text{diag}(\alpha_I^l)H_I^l$$

$$(11)$$

The framework with option 1 in step 1 is called ACM framework and with option 2 is named ACMII framework. In step 1, ACM-GCN and ACMII-GCN implement distinct feature extraction for 3 channels by a set of filterbank and 3 filtered components $H_L^l, H_H^l, H_I^l$ are obtained. To adaptively exploit information from each channel, ACM-GCN and ACMII-GCN first extract nonlinear information from the filtered signals, then use $W_{\text{Mix}}^l$ to learn which channel is important or not for each node, leading to the row-wise weight vectors $\alpha_L^l, \alpha_H^l, \alpha_I^l \in \mathbb{R}^{N \times 1}$ whose $i$-th elements are the weights for the $i$-th node. These three vectors are then used as weights in defining the updated $H^l$ in step 3. See Appendix F for the performance comparison with basline models on synthetic datasets.

**Complexity** Number of learnable parameters in layer $l$ of ACM-GCN and ACMII-GCN is $3F_{l-1}(F_l + 1) + 9$, while it is $F_{l-1}F_l$ in GCN. The computation of step 1-3 takes $NF_l(8 + 6F_{l-1}) + 2F_l(\text{nnz}(H_{\text{LP}}) + \text{nnz}(H_{\text{HP}})) + 18N$ flops, while GCN layer takes $2NF_{l-1}F_l + 2F_l(\text{nnz}(H_{\text{LP}}))$ flops, where $\text{nnz}(\cdot)$ is the number of non-zero elements. An ablation study and a detailed comparison on running time is conducted in section 6.1.

**Limitations of Diversification Operation** Just like any other method, Diversification operation may not work well in all harmful heterophily cases. For example, when we have more than 2 classes and consider an imbalanced dataset where several small clusters with distinctive labels are densely connected to a large cluster. In this case, the surrounding differences of nodes in small clusters are similar, *i.e.,* the neighborhood differences are mainly from their connection to the same large cluster, and this possibly makes diversification operation fail to discriminate them. See a more detailed demonstration and discussion in Appendix G.

## 5 PRIOR WORK

We discuss relevant work of GNNs on addressing heterophily challenge in this part. Authors in (Abu-El-Haija et al., 2019) acknowledge the difficulty of learning on graphs with weak homophily and propose MixHop to extract features from multi-hop neighborhood to get more information. Authors in (Hou et al., 2019) propose measurements based on feature smoothness and label smoothness that are potentially helpful to guide GNNs on dealing with heterophilous graphs. Geom-GCN (Pei et al., 2020) precomputes unsupervised node embeddings and uses graph structure defined by geometric relationships in the embedding space to define the bi-level aggregation process to handle heterophily. $H_2$GCN (Zhu et al., 2020b) combines 3 key designs to address heterophily: (1) ego- and neighbor-embedding separation; (2) higher-order neighborhoods; (3) combination of intermediate representations. CPGNN (Zhu et al., 2020a) models label correlations by the compatibility matrix, which is beneficial for heterophily settings, and propagates a prior belief estimation into GNNs by the compatibility matrix. FBGNN (Luan et al., 2020b) first proposes to use filterbank to address heterophily problem, but it does not fully explain the insights behind HP filters and does not contain identity channel and node-wise channel mixing mechanism. FAGCN (Bo et al., 2021) learns edge-level aggregation weights as GAT (Velickovic et al., 2017) but allows the weights to be negative which enables the network to capture the high-frequency components in graph signals. GPRGNN (Chien et al., 2021) uses learnable weights that can be both positive and negative for feature propagation, it allows GPRGNN to adapt heterophily structure of graph and is able to handle both high- and low-frequency parts of the graph signals. (See Appendix J for a more comprehensive comparison between ACM-GNNs, ACMII-GNNs and FAGCN, GPRGNN.)

# 6 EXPERIMENTS ON REAL-WORLD DATASETS

In this section, we evaluate ACM and ACMII framework on real-world datasets. We first conduct ablation studies in subsection 6.1 to validate the effectiveness of different components. Then, we compare with the state-of-the-arts (SOTA) models in subsection 6.2. The hyperparameter searching range and computing resources for all experiments are attached in Appendix A.

## 6.1 ABLATION STUDY & EFFICIENCY

| Ablation Study on Different Components in ACM-SGC and ACM-GCN (%) | | | | | | | | | | | | | | |
|---|---|---|---|---|---|---|---|---|---|---|---|---|---|---|
| Baseline Models | LP | HP | Identity | Mixing | Cornell Acc ± Std | Wisconsin Acc ± Std | Texas Acc ± Std | Film Acc ± Std | Chameleon Acc ± Std | Squirrel Acc ± Std | Cora Acc ± Std | CiteSeer Acc ± Std | PubMed Acc ± Std | Rank |
| ACM-SGC-1 w/ | ✓ | | | | 70.98 ± 8.39 | 70.38 ± 2.85 | 83.28 ± 5.43 | 25.26 ± 1.18 | 64.86 ± 1.81 | 47.62 ± 1.27 | 85.12 ± 1.64 | 79.66 ± 0.75 | 85.5 ± 0.76 | 12.89 |
| | ✓ | ✓ | | ✓ | 83.28 ± 5.81 | 91.88 ± 1.61 | 90.98 ± 2.46 | 36.76 ± 1.01 | 65.27 ± 1.9 | 47.27 ± 1.37 | 86.8 ± 1.08 | 80.98 ± 1.68 | 87.21 ± 0.42 | 10.44 |
| | ✓ | | ✓ | ✓ | 93.93 ± 3.6 | 95.25 ± 1.84 | 93.93 ± 2.54 | 38.38 ± 1.13 | 63.83 ± 2.07 | 46.79 ± 0.75 | 86.73 ± 1.28 | 80.57 ± 0.99 | 87.8 ± 0.58 | 9.44 |
| | ✓ | ✓ | ✓ | | 88.2 ± 4.39 | 93.5 ± 2.95 | 92.95 ± 2.94 | 37.19 ± 0.87 | 62.82 ± 1.84 | 44.94 ± 0.93 | 85.22 ± 1.35 | 80.75 ± 1.68 | 88.11 ± 0.21 | 11.00 |
| | ✓ | ✓ | ✓ | ✓ | 93.77 ± 1.91 | 93.25 ± 2.92 | 93.61 ± 1.55 | 39.33 ± 1.25 | 63.68 ± 1.62 | 46.4 ± 1.13 | 86.63 ± 1.13 | 80.96 ± 0.93 | 87.75 ± 0.88 | 10.00 |
| ACM-GCN w/ | ✓ | | | | 82.46 ± 3.11 | 75.5 ± 2.92 | 83.11 ± 3.2 | 35.51 ± 0.99 | 64.18 ± 2.62 | 44.76 ± 1.39 | 87.78 ± 0.96 | 81.39 ± 1.23 | 88.9 ± 0.32 | 11.44 |
| | ✓ | ✓ | | ✓ | 82.13 ± 2.59 | 86.62 ± 4.61 | 89.19 ± 3.04 | 38.06 ± 1.35 | **69.21 ± 1.68** | 57.2 ± 1.01 | 88.93 ± 1.55 | **81.96 ± 0.91** | 90.01 ± 0.8 | 7.22 |
| | ✓ | | ✓ | ✓ | 94.26 ± 2.23 | 96.13 ± 2.2 | 94.1 ± 2.95 | 41.51 ± 0.99 | 67.44 ± 2.14 | 53.97 ± 1.39 | 88.95 ± 0.9 | 81.72 ± 1.22 | 90.88 ± 0.55 | 4.44 |
| | ✓ | ✓ | ✓ | | 91.64 ± 2 | 95.37 ± 3.31 | **95.25 ± 2.37** | 40.47 ± 1.49 | 68.93 ± 2.04 | 54.78 ± 1.27 | **89.13 ± 1.77** | **81.96 ± 2.03** | **91.01 ± 0.7** | 3.11 |
| | ✓ | ✓ | ✓ | ✓ | 94.75 ± 2.62 | **96.75 ± 1.6** | 95.08 ± 3.2 | 41.62 ± 1.15 | 69.04 ± 1.74 | **58.02 ± 1.86** | 88.95 ± 1.3 | 81.80 ± 1.26 | 90.69 ± 0.53 | 2.78 |
| ACMII-GCN w/ | ✓ | ✓ | | ✓ | 82.46 ± 3.03 | 91.00 ± 1.75 | 90.33 ± 2.69 | 38.39 ± 0.75 | 67.59 ± 2.14 | 53.67 ± 1.71 | **89.13 ± 1.14** | 81.75 ± 0.85 | 89.87 ± 0.39 | 7.44 |
| | ✓ | | ✓ | ✓ | 94.26 ± 2.57 | 96.00 ± 2.15 | 94.26 ± 2.96 | 40.96 ± 1.2 | 66.35 ± 1.76 | 50.78 ± 2.07 | 89.06 ± 1.07 | 81.86 ± 1.22 | 90.71 ± 0.67 | 4.67 |
| | ✓ | ✓ | ✓ | | 91.48 ± 1.43 | 96.25 ± 2.09 | 93.77 ± 2.91 | 40.27 ± 1.07 | 66.52 ± 2.65 | 52.9 ± 1.64 | 88.83 ± 1.16 | 81.54 ± 0.95 | 90.6 ± 0.47 | 6.67 |
| | ✓ | ✓ | ✓ | ✓ | **95.9 ± 1.83** | 96.62 ± 2.44 | 95.25 ± 3.15 | **41.84 ± 1.15** | 68.38 ± 1.36 | 54.53 ± 2.09 | 89.00 ± 0.72 | 81.79 ± 0.95 | 90.74 ± 0.5 | 2.78 |
| Comparison of Average Running Time Per Epoch(ms) | | | | | | | | | | | | | | |
| ACM-SGC-1 w/ | ✓ | | | | 2.53 | 2.83 | 2.5 | 3.18 | 3.48 | 4.65 | 3.47 | 3.43 | 4.04 | |
| | ✓ | ✓ | | ✓ | 4.01 | 4.57 | 4.24 | 4.55 | 4.76 | 5.09 | 5.39 | 4.69 | 4.75 | |
| | ✓ | | ✓ | ✓ | 3.88 | 4.01 | 4.04 | 4.43 | 4.06 | 4.5 | 4.38 | 3.82 | 4.16 | |
| | ✓ | ✓ | ✓ | | 3.31 | 3.49 | 3.18 | 3.7 | 3.53 | 4.83 | 3.92 | 3.87 | 4.24 | |
| | ✓ | ✓ | ✓ | ✓ | 5.53 | 5.96 | 5.43 | 5.21 | 5.41 | 6.96 | 6 | 5.9 | 6.04 | |
| ACM-GCN w/ | ✓ | | | | 3.67 | 3.74 | 3.59 | 4.86 | 4.96 | 6.41 | 4.24 | 4.18 | 5.08 | |
| | ✓ | ✓ | | ✓ | 6.63 | 8.06 | 7.89 | 8.11 | 7.8 | 9.39 | 7.82 | 7.38 | 8.74 | |
| | ✓ | | ✓ | ✓ | 5.73 | 5.91 | 5.93 | 6.86 | 6.35 | 7.15 | 7.34 | 6.65 | 6.8 | |
| | ✓ | ✓ | ✓ | | 5.16 | 5.25 | 5.2 | 5.93 | 5.64 | 8.02 | 5.73 | 5.65 | 6.16 | |
| | ✓ | ✓ | ✓ | ✓ | 8.25 | 8.11 | 7.89 | 7.97 | 8.41 | 11.9 | 8.84 | 8.38 | 8.63 | |
| ACMII-GCN w/ | ✓ | ✓ | | ✓ | 6.62 | 7.35 | 7.39 | 7.62 | 7.33 | 9.69 | 7.49 | 7.58 | 7.97 | |
| | ✓ | | ✓ | ✓ | 6.3 | 6.05 | 6.26 | 6.87 | 6.44 | 6.5 | 6.14 | 7.21 | 6.6 | |
| | ✓ | ✓ | ✓ | | 5.24 | 5.27 | 5.46 | 5.72 | 5.65 | 7.87 | 5.48 | 5.65 | 6.33 | |
| | ✓ | ✓ | ✓ | ✓ | 7.59 | 8.28 | 8.06 | 8.85 | 8 | 10 | 8.27 | 8.5 | 8.68 | |

Table 1: Ablation study on 9 real-world datasets Pei et al. (2020). Cell with ✓ means the component is applied to the baseline model. The best test results are highlighted.

We investigate the effectiveness and efficiency of adding HP, identity channels and the adaptive mixing mechanism in the proposed framework by ablation study. Specifically, we apply the components of ACM to SGC-1 (Wu et al., 2019) [6] and the components of ACM and ACMII to GCN (Kipf & Welling, 2016) separately. We run 10 times on each dataset with the same 60%/20%/20% random splits for train/validation/test used in (Chien et al., 2021) and report the average test accuracy as well as the standard deviation. We also record the average running time per epoch (in milliseconds) to compare the efficiency. We set the temperature $T$ in equation 11 to be 3.

From the results we can see that on most datasets, the additional HP and identity channels are helpful, even on strong homophily datasets, such as *Cora, CiteSeer and PubMed*. The adaptive mixing mechanism also shows its advantage over the method that directly adds the three channels together. This illustrates the necessity of learning to customize the channel usage adaptively for different nodes. As for efficiency, we can see that the running time is approximately doubled under ACM and ACMII framework than the original model.

## 6.2 COMPARISON WITH STATE-OF-THE-ART MODELS

**Datasets & Experimental Setup** In this section, we implement SGC (Wu et al., 2019) with 1 hop and 2 hops (SGC-1, SGC-2), GCNII (Chen et al., 2020), GCNII* (Chen et al., 2020), GCN (Kipf & Welling, 2016) and snowball networks with 2 and 3 layers (snowball-2, snowball-3) and apply them

---

[6]We only test ACM-SGC-1 because SGC-1 does not contrain any non-linearity and ACM-SGC-1 and ACMII-SGC-1 are the same.

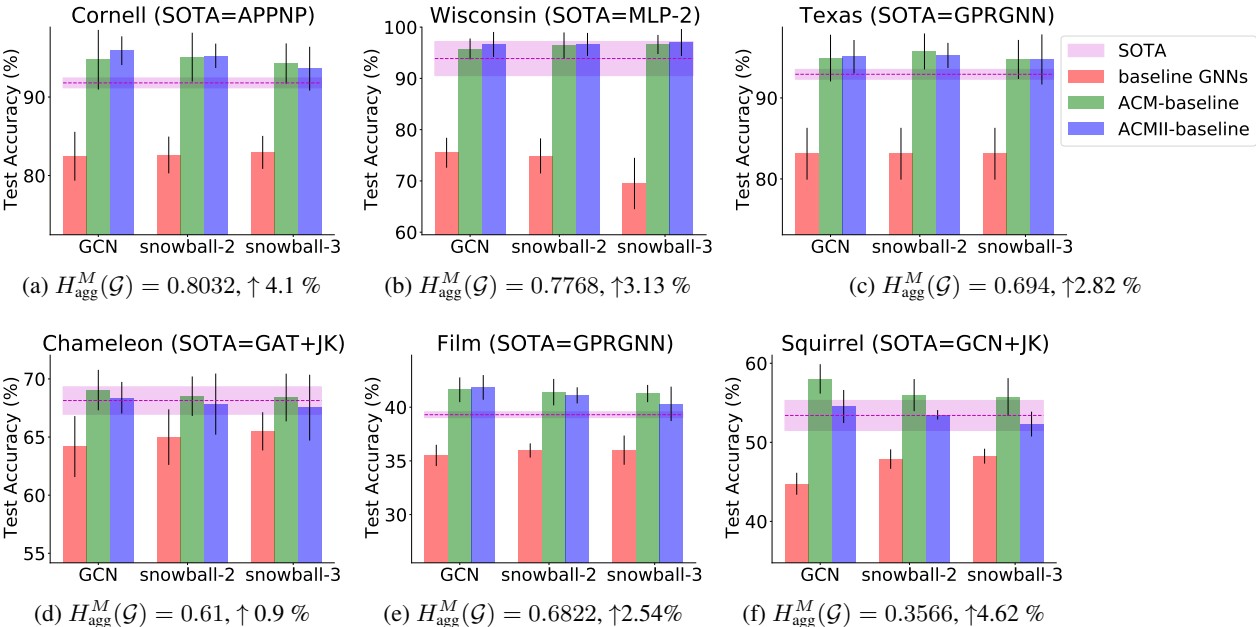

Figure 4: Comparison of selected baseline GNNs (red), ACM-baseline (green), ACMII-baseline (blue) with SOTA (magenta) models on 6 selected datasets. The black line and the error bar indicate the standard deviation. The symbol "↑" means the amount of improvement of the best ACM-baseline and ACM-baseline over the SOTA models. See Appendix H for a detailed discussion of the relation between $H_{agg}^M$ and the performance of GNNs.

in ACM or ACMII framework[7]: we use $\hat{A}_{rw}$ as LP filter and the corresponding HP filter is $I - \hat{A}_{rw}$ and they are deterministic. We compare them with several baseline and SOTA GNN models: MLP with 2 layers (MLP-2), GAT (Velickovic et al., 2017), APPNP (Klicpera et al., 2018), GPRGNN (Chien et al., 2021), $H_2$GCN (Zhu et al., 2020b), MixHop (Abu-El-Haija et al., 2019), GCN+JK (Kipf & Welling, 2016; Xu et al., 2018; Lim et al., 2021), GAT+JK (Velickovic et al., 2017; Xu et al., 2018; Lim et al., 2021), FAGCN (Bo et al., 2021) GraphSAGE (Hamilton et al., 2017) and Geom-GCN (Pei et al., 2020). Besides the 9 benchmark datasets *Cornell*, *Wisconsin*, *Texas*, *Film*, *Chameleon*, *Squirrel*, *Cora*, *Citeseer* and *Pubmed* used in (Rozemberczki et al., 2021; Pei et al., 2020), we further test the above models on a new benchmark dataset, *Deezer-Europe*, that is proposed in (Rozemberczki & Sarkar, 2020). On each dataset used in (Rozemberczki et al., 2021; Pei et al., 2020), we test the models 10 times following the same early stopping strategy, the same random data splitting method [8] and Adam (Kingma & Ba, 2014) optimizer as used in GPRGNN (Chien et al., 2021). For *Deezer-Europe*, we test the above models 5 times with the same early stopping strategy, the same fixed splits and AdamW (Loshchilov & Hutter, 2017) used in (Lim et al., 2021).

To better visualize the performance boost and the comparison with SOTA models, in Figure 4, we plot the bar charts of the test accuracy of SOTA models, 3 selected baselines (GCN, snowball-2, snowball-3) and their ACM and ACMII augmented models on 6 most commonly used benchmark heterophily datasets (See table 4 in Appendix A.3 for the full results and comparison). We can see that after being applied in ACM or ACMII framework, the performance of the 3 baseline models are significantly boosted on all tasks and can achieve SOTA performance. Especially on *Cornell, Texas, Film* and *Squirrel*, the augmented models significantly outperform the current SOTA models. Overall, It suggests that ACM or ACMII framework can help GNNs to generalize better on node classification tasks on heterophilous graphs.

---

[7]GCNII and GCNII* are hard to be implemented under ACMII framework. See Appendix A.6 for explanation.

[8]See table 9 in Appendix I for the performance comparison with several SOTA models on the fixed 48%/32%/20% splits provided by (Pei et al., 2020).

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

## A  Hyperparameters & Details of The Experiments

### A.1  Hyperparameter Searching Range for Synthetic Experiments

| Models\Hyperparameters | lr | weight_decay | dropout | hidden |
|---|---|---|---|---|
| MLP-1 | 0.05 | {5e-5, 1e-4, 5e-4, 1e-3, 5e-3 } | - | - |
| SGC-1 | 0.05 | {5e-5, 1e-4, 5e-4, 1e-3, 5e-3} | - | - |
| ACM-SGC-1 | 0.05 | {5e-5, 1e-4, 5e-4, 1e-3, 5e-3} | { 0.1, 0.3, 0.5, 0.7, 0.9} | - |
| MLP-2 | 0.05 | {5e-5, 1e-4, 5e-4, 1e-3, 5e-3} | { 0.1, 0.3, 0.5, 0.7, 0.9} | 64 |
| GCN | 0.05 | {5e-5, 1e-4, 5e-4, 1e-3, 5e-3} | { 0.1, 0.3, 0.5, 0.7, 0.9} | 64 |
| ACM-GCN | 0.05 | {5e-5, 1e-4, 5e-4, 1e-3, 5e-3} | { 0.1, 0.3, 0.5, 0.7, 0.9} | 64 |

Table 2: Hyperparameter Searching Range for Synthetic Experiments

### A.2  Hyperparameter Searching Range for Ablation Study

| Models\Hyperparameters | lr | weight_decay | dropout | hidden |
|---|---|---|---|---|
| SGC-LP+HP | {0.01, 0.05, 0.1} | {0, 5e-6, 1e-5, 5e-5, 1e-4, 5e-4, 1e-3, 5e-3, 1e-2} | - | - |
| SGC-LP+Identity | {0.01, 0.05, 0.1} | {0, 5e-6, 1e-5, 5e-5, 1e-4, 5e-4, 1e-3, 5e-3, 1e-2} | - | - |
| ACM-SGC-no adaptive mixing | {0.01, 0.05, 0.1} | {0, 5e-6, 1e-5, 5e-5, 1e-4, 5e-4, 1e-3, 5e-3, 1e-2} | {0, 0.1, 0.2, 0.3, 0.4, 0.5, 0.6, 0.7,0.8,0.9} | - |
| GCN-LP+HP | {0.01, 0.05, 0.1} | {0, 5e-6, 1e-5, 5e-5, 1e-4, 5e-4, 1e-3, 5e-3, 1e-2} | {0, 0.1, 0.2, 0.3, 0.4, 0.5, 0.6, 0.7,0.8,0.9} | 64 |
| GCN-LP+Identity | {0.01, 0.05, 0.1} | {0, 5e-6, 1e-5, 5e-5, 1e-4, 5e-4, 1e-3, 5e-3, 1e-2} | {0, 0.1, 0.2, 0.3, 0.4, 0.5, 0.6, 0.7,0.8,0.9} | 64 |
| ACM-GCN-no adaptive mixing | {0.01, 0.05, 0.1} | {0, 5e-6, 1e-5, 5e-5, 1e-4, 5e-4, 1e-3, 5e-3, 1e-2} | {0, 0.1, 0.2, 0.3, 0.4, 0.5, 0.6, 0.7,0.8,0.9} | 64 |

Table 3: Hyperparameter Searching Range for Ablation Study

### A.3  Full Results of The Comparison with SOTA Models

The main results of the full sets of experiments [9] with statistics of datasets are summarized in Table 4, where we report the mean accuracy and standard deviation. We can see that after applied in ACM or ACMII framework, the performance of baseline models are boosted on almost all tasks and achieve SOTA performance on 8 out of 10 datasets. Especially, ACMII-GCN performs the best in terms of average rank (3.40) across all datasets. Overall, It suggests that ACM or ACMII framework can significantly increase the performance of GNNs on node classification tasks on heterophilous graphs.

### A.4  Hyperparameter Searching Range for GNNs on Real-world Datasets

See table 5 for the hyperparameter seaching range of baseline GNNs, ACM-GNNs, ACMII-GNNs and several SOTA models.

### A.5  Optimal Hyperparameters for Baselines and ACM(II)-GNNs on Real-world Tasks

See the reported optimal hyperparameters for baseline GNNs in table 6 and for ACM-GNNs and ACMII-GNNs in table 7.

---

[9]The splits for all these experiments are random 60%/20%/20% splits for train/valid/test. The open source code we use is from https://github.com/jianhao2016/GPRGNN/blob/f4aaad6ca28c83d3121338a4c4fe5d162edfa9a2/src/utils.py#L16. See table 9 in Appendix I for the performance comparison with several SOTA models on the fixed 48%/32%/20% splits provided by (Pei et al., 2020).

| | Cornell | Wisconsin | Texas | Film | Chameleon | Squirrel | Deezer-Europe | Cora | CiteSeer | PubMed | |
|---|---|---|---|---|---|---|---|---|---|---|---|
| #nodes | 183 | 251 | 183 | 7,600 | 2,277 | 5,201 | 28,281 | 2,708 | 3,327 | 19,717 | |
| #edges | 295 | 499 | 309 | 33,544 | 36,101 | 217,073 | 92,752 | 5,429 | 4,732 | 44,338 | |
| #features | 1,703 | 1,703 | 1,703 | 931 | 2,325 | 2,089 | 31,241 | 1,433 | 3,703 | 500 | |
| #classes | 5 | 5 | 5 | 5 | 5 | 5 | 2 | 7 | 6 | 3 | |
| $H_{edge}$ | 0.5669 | 0.4480 | 0.4106 | 0.3750 | 0.2795 | 0.2416 | 0.5251 | 0.8100 | 0.7362 | 0.8024 | |
| $H_{node}$ | 0.3855 | 0.1498 | 0.0968 | 0.2210 | 0.2470 | 0.2156 | 0.5299 | 0.8252 | 0.7175 | 0.7924 | |
| $H_{class}$ | 0.0468 | 0.0941 | 0.0013 | 0.0110 | 0.0620 | 0.0254 | 0.0304 | 0.7657 | 0.6270 | 0.6641 | |
| Data Splits | 60%/20%/20% | 60%/20%/20% | 60%/20%/20% | 60%/20%/20% | 60%/20%/20% | 60%/20%/20% | 50%/25%/25% | 60%/20%/20% | 60%/20%/20% | 60%/20%/20% | |
| $H_{agg}^{M}(\mathcal{G})$ | 0.8032 | 0.7768 | 0.694 | 0.6822 | 0.61 | 0.3566 | 0.5790 | 0.9904 | 0.9826 | 0.9432 | |

| | Test Accuracy (%) of State-of-the-art Models, Baseline GNN Models and ACM-GNN models | | | | | | | | | | Rank |
|---|---|---|---|---|---|---|---|---|---|---|---|
| MLP-2* | $91.30 \pm 0.70$ | $93.87 \pm 3.33$ | $92.26 \pm 0.71$ | $38.58 \pm 0.25$ | $46.72 \pm 0.46$ | $31.28 \pm 0.27$ | $66.55 \pm 0.72$ | $76.44 \pm 0.30$ | $76.25 \pm 0.28$ | $86.43 \pm 0.13$ | 18.60 |
| GAT* | $76.00 \pm 1.01$ | $71.01 \pm 4.66$ | $78.87 \pm 0.86$ | $35.98 \pm 0.23$ | $63.9 \pm 0.46$ | $42.72 \pm 0.33$ | $61.09 \pm 0.77$ | $76.70 \pm 0.42$ | $67.20 \pm 0.46$ | $83.28 \pm 0.12$ | 21.40 |
| APPNP* | $91.80 \pm 0.63$ | $92.00 \pm 3.59$ | $91.18 \pm 0.70$ | $38.86 \pm 0.24$ | $51.91 \pm 0.56$ | $34.77 \pm 0.34$ | $67.21 \pm 0.56$ | $79.41 \pm 0.38$ | $68.59 \pm 0.30$ | $85.02 \pm 0.09$ | 18.00 |
| GPRGNN* | $91.36 \pm 0.70$ | $93.75 \pm 2.37$ | $92.92 \pm 0.61$ | $39.30 \pm 0.27$ | $67.48 \pm 0.40$ | $49.93 \pm 0.53$ | $66.90 \pm 0.50$ | $79.51pm 0.36$ | $67.63 \pm 0.38$ | $85.07 \pm 0.09$ | 14.40 |
| H2GCN | $86.23 \pm 4.71$ | $87.5 \pm 1.77$ | $85.90 \pm 3.53$ | $38.85 \pm 1.17$ | $52.30 \pm 0.48$ | $30.39 \pm 1.22$ | $\mathbf{67.22 \pm 0.90}$ | $87.52 \pm 0.61$ | $79.97 \pm 0.69$ | $87.78 \pm 0.28$ | 17.00 |
| MixHop | $60.33 \pm 28.53$ | $77.25 \pm 7.80$ | $76.39 \pm 7.66$ | $33.13 \pm 2.40$ | $36.28 \pm 10.22$ | $24.55 \pm 2.60$ | $66.80 \pm 0.58$ | $65.65 \pm 11.31$ | $49.52 \pm 13.35$ | $87.04 \pm 4.10$ | 23.50 |
| GCN+JK | $66.56 \pm 13.82$ | $62.50 \pm 15.75$ | $80.66 \pm 1.91$ | $32.72 \pm 2.62$ | $64.68 \pm 2.85$ | $53.40 \pm 1.90$ | $60.99 \pm 0.14$ | $86.90 \pm 1.51$ | $73.77 \pm 1.85$ | $90.09 \pm 0.68$ | 18.80 |
| GAT+JK | $74.43 \pm 10.24$ | $69.50 \pm 3.12$ | $75.41 \pm 7.18$ | $35.41 \pm 0.97$ | $68.14 \pm 1.18$ | $52.28 \pm 3.61$ | $59.66 \pm 0.92$ | $89.52 \pm 0.43$ | $74.49 \pm 2.76$ | $89.15 \pm 0.87$ | 16.70 |
| FAGCN | $88.03 \pm 5.6$ | $89.75 \pm 6.37$ | $88.85 \pm 4.39$ | $31.59 \pm 1.37$ | $49.47 \pm 2.84$ | $42.24 \pm 1.2$ | $66.86 p, 0.53$ | $88.85 \pm 1.36$ | $\mathbf{82.37 \pm 1.46}$ | $89.98 \pm 0.54$ | 14.10 |
| GraphSAGE | $71.41 \pm 1.24$ | $64.85 \pm 5.14$ | $79.03 \pm 1.20$ | $36.37 \pm 0.21$ | $62.15 \pm 0.42$ | $41.26 \pm 0.26$ | OOM | $86.58 \pm 0.26$ | $78.24 \pm 0.30$ | $86.85 \pm 0.11$ | 20.89 |
| Geom-GCN† | 60.81 | 64.12 | 67.57 | 31.63 | 60.9 | 38.14 | NA | 85.27 | 77.99 | 90.05 | 22.67 |
| SGC-1 | $70.98 \pm 8.39$ | $70.38 \pm 2.85$ | $83.28 \pm 5.43$ | $25.26 \pm 1.18$ | $64.86 \pm 1.81$ | $47.62 \pm 1.27$ | $59.73 \pm 0.12$ | $85.12 \pm 1.64$ | $79.66 \pm 0.75$ | $85.5 \pm 0.76$ | 20.10 |
| SGC-2 | $72.62 \pm 9.92$ | $74.75 \pm 2.89$ | $81.31 \pm 3.3$ | $28.81 \pm 1.11$ | $62.67 \pm 2.41$ | $41.25 \pm 1.4$ | $61.56 \pm 0.51$ | $85.48 \pm 1.48$ | $80.75 \pm 1.15$ | $85.36 \pm 0.52$ | 20.70 |
| GCNII | $89.18 \pm 3.96$ | $83.25 \pm 2.69$ | $82.46 \pm 4.58$ | $40.82 \pm 1.79$ | $60.35 \pm 2.7$ | $38.81 \pm 1.97$ | $66.38 \pm 0.45$ | $88.98 \pm 1.33$ | $81.58 \pm 1.3$ | $89.8 \pm 0.3$ | 14.80 |
| GCNII* | $90.49 \pm 4.45$ | $89.12 \pm 3.06$ | $88.52 \pm 3.02$ | $41.54 \pm 0.99$ | $62.8 \pm 2.87$ | $38.31 \pm 1.3$ | $66.42 \pm 0.56$ | $88.93 \pm 1.37$ | $81.83 \pm 1.78$ | $89.98 \pm 0.52$ | 12.30 |
| GCN | $82.46 \pm 3.11$ | $75.5 \pm 2.92$ | $83.11 \pm 3.2$ | $35.51 \pm 0.99$ | $64.18 \pm 2.62$ | $44.76 \pm 1.39$ | $62.23 \pm 0.53$ | $87.78 \pm 0.96$ | $81.39 \pm 1.23$ | $88.9 \pm 0.32$ | 16.30 |
| Snowball-2 | $82.62 \pm 2.34$ | $74.88 \pm 3.42$ | $83.11 \pm 3.2$ | $35.97 \pm 0.66$ | $64.99 \pm 2.39$ | $47.88 \pm 1.23$ | OOM | $88.64 \pm 1.15$ | $81.53 \pm 1.71$ | $89.04 \pm 0.49$ | 15.22 |
| Snowball-3 | $82.95 \pm 2.1$ | $69.5 \pm 5.01$ | $83.11 \pm 3.2$ | $36.00 \pm 1.36$ | $65.49 \pm 1.64$ | $48.25 \pm 0.94$ | OOM | $89.33 \pm 1.3$ | $80.93 \pm 1.32$ | $88.8 \pm 0.82$ | 14.78 |
| ACM-SGC-1 | $93.77 \pm 1.91$ | $93.25 \pm 2.92$ | $93.61 \pm 1.55$ | $39.33 \pm 1.25$ | $63.68 \pm 1.62$ | $46.4 \pm 1.13$ | $66.67 \pm 0.56$ | $86.63 \pm 1.13$ | $80.96 \pm 0.93$ | $87.75 \pm 0.88$ | 12.60 |
| ACM-SGC-2 | $93.77 \pm 2.17$ | $94.00 \pm 2.61$ | $93.44 \pm 2.54$ | $40.13 \pm 1.21$ | $60.48 \pm 1.55$ | $40.91 \pm 1.39$ | $66.53 \pm 0.57$ | $87.64 \pm 0.99$ | $80.93 \pm 1.16$ | $88.79 \pm 0.5$ | 13.40 |
| ACM-GCNII | $92.62 \pm 3.13$ | $94.63 \pm 2.96$ | $92.46 \pm 1.97$ | $41.37 \pm 1.37$ | $58.73 \pm 2.52$ | $40.9 \pm 1.58$ | $66.39 \pm 0.56$ | $89.1 \pm 1.61$ | $82.28 \pm 1.12$ | $90.12 \pm 0.4$ | 10.40 |
| ACM-GCNII* | $93.44 \pm 2.74$ | $94.37 \pm 2.81$ | $93.28 \pm 2.79$ | $41.27 \pm 1.24$ | $61.66 \pm 2.29$ | $38.32 \pm 1.5$ | $66.6 \pm 0.57$ | $89.00 \pm 1.35$ | $81.69 \pm 1.25$ | $90.18 0.51$ | 10.10 |
| ACM-GCN | $94.75 \pm 3.8$ | $95.75 \pm 2.03$ | $94.92 \pm 2.88$ | $41.62 \pm 1.15$ | $\mathbf{69.04 \pm 1.74}$ | $\mathbf{58.02 \pm 1.86}$ | $67.01 \pm 0.38$ | $88.62 \pm 1.22$ | $81.68 \pm 0.97$ | $90.66 \pm 0.47$ | 4.80 |
| ACM-Snowball-2 | $95.08 \pm 3.11$ | $96.38 \pm 2.59$ | $\mathbf{95.74 \pm 2.22}$ | $41.4 \pm 1.23$ | $68.51 \pm 1.7$ | $55.97 \pm 2.03$ | OOM | $88.83 \pm 1.49$ | $81.58 \pm 1.23$ | $90.81 \pm 0.52$ | 4.44 |
| ACM-Snowball-3 | $94.26 \pm 2.57$ | $96.62 \pm 1.86$ | $94.75 \pm 2.41$ | $41.27 \pm 0.8$ | $68.4 \pm 2.05$ | $55.73 \pm 2.39$ | OOM | $\mathbf{89.59 \pm 1.58}$ | $81.32 \pm 0.97$ | $\mathbf{91.44 \pm 0.59}$ | 4.44 |
| ACMII-GCN | $\mathbf{95.9 \pm 1.83}$ | $96.62 \pm 2.44$ | $95.08 \pm 2.07$ | $\mathbf{41.84 \pm 1.15}$ | $68.38 \pm 1.36$ | $54.53 \pm 2.09$ | $67.15 \pm 0.41$ | $89.00 \pm 0.72$ | $81.79 \pm 0.95$ | $90.74 \pm 0.5$ | **3.40** |
| ACMII-Snowball-2 | $95.25 \pm 1.55$ | $96.63 \pm 2.24$ | $95.25 \pm 1.55$ | $41.1 \pm 0.75$ | $67.83 \pm 2.63$ | $53.48 \pm 0.6$ | OOM | $88.95 \pm 1.04$ | $82.07 \pm 1.04$ | $90.56 \pm 0.39$ | 4.78 |
| ACMII-Snowball-3 | $93.61 \pm 2.79$ | $\mathbf{97.00 \pm 2.63}$ | $94.75 \pm 3.09$ | $40.31 \pm 1.6$ | $67.53 \pm 2.83$ | $52.31 \pm 1.57$ | OOM | $89.36 \pm 1.26$ | $81.56 \pm 1.15$ | $91.31 \pm 0.6$ | 5.89 |

Table 4: Experimental results: average test accuracy $\pm$ standard deviation on 10 real-world benchmark datasets. The best results are highlighted. Results "*" are reported from Chien et al. (2021); Lim et al. (2021) and results "†" are from Pei et al. (2020). NA means the reported results are not available and OOM means out of memory.

| Models\Hyperparameters | lr | weight_decay | dropout | hidden | lambda | alpha_l | head | layers | JK type |
|---|---|---|---|---|---|---|---|---|---|
| H2GCN | 0.01 | 0.001 | {0, 0.5} | {8, 16, 32, 64} | - | - | - | {1, 2} | - |
| MixHop | 0.01 | 0.001 | 0.5 | {8, 16, 32} | - | - | - | {2, 3} | - |
| GCN+JK | {0.1, 0.01, 0.001} | 0.001 | 0.5 | {4, 8, 16, 32, 64} | - | - | - | 2 | {max, cat} |
| GAT+JK | {0.1, 0.01, 0.001} | 0.001 | 0.5 | {4, 8, 12, 32} | - | - | {2,4,8} | 2 | {max, cat} |
| GCNII, GCNII* | 0.01 | {0, 5e-6, 1e-5, 5e-5, 1e-4, 5e-4, 1e-3} for Deezer-Europe and {0, 5e-6, 1e-5, 5e-5, 1e-4, 5e-4, 1e-3, 5e-3, 1e-2} for others | 0.5 | 64 | {0.5, 1, 1.5} | {0.1,0.2,0.3,0,4,0.5} | - | {4, 8, 16, 32} for Deezer-Europe and {4, 8, 16, 32, 64} for others | - |
| Baselines: {SGC-1, SGC-2, GCN, Snowball-2, Snowball-3, FAGCN}; ACM-{SGC-1, SGC-2, GCN, Snowball-2, Snowball-3}; ACMII-{SGC-1, SGC-2, GCN, Snowball-2, Snowball-3} | {0.002, 0.01, 0.05} for Deezer-Europe and {0.01, 0.05, 0.1} for others | {0, 5e-6, 1e-5, 5e-5, 1e-4, 5e-4, 1e-3} for Deezer-Europe and {0, 5e-6, 1e-5, 5e-5, 1e-4, 5e-4, 1e-3, 5e-3, 1e-2} for others | {0, 0.1, 0.2, 0.3, 0.4, 0.5, 0.6, 0.7, 0.8, 0.9} | 64 | - | - | - | - | - |
| GraphSAGE | {0.01,0.05, 0.1} | {0, 5e-6, 1e-5, 5e-5, 1e-4, 5e-4, 1e-3} for Deezer-Europe and {0, 5e-6, 1e-5, 5e-5, 1e-4, 5e-4, 1e-3, 5e-3, 1e-2} for others | { 0, 0.1, 0.2, 0.3, 0.4, 0.5, 0.6, 0.7, 0.8, 0.9} | 8 for Deezer-Europe and 64 for others | - | - | - | - | - |
| ACM-{GCNII, GCNII*} | 0.01 | {0, 5e-6, 1e-5, 5e-5, 1e-4, 5e-4, 1e-3} for Deezer-Europe and {0, 5e-6, 1e-5, 5e-5, 1e-4, 5e-4, 1e-3, 5e-3, 1e-2} for others | { 0, 0.1, 0.2, 0.3, 0.4, 0.5, 0.6, 0.7, 0.8, 0.9} | 64 | - | - | - | {1,2,3,4} | - |

Table 5: Hyperparameter Searching Range for Real-world Datasets

## A.6 Details of the Implementation of ACM and ACMII Frameworks

Unlike some baseline GNN models, in ACM(II) framework, we first use dropout operation over the input data. The implementation of ACM(II)-GCN and ACM(II)-snowball is straightforward, but SGC-1, SGC-2, GCNII and GCNII* are not able to be applied under ACM(II) framework and we will make an explanation as follows.

- SGC-1 and SGC-2: SGC does not contain nonlinearity, so the option 1 and option 2 in step 1 of equation 11 is the same for ACM-SGC and ACMII-SGC. Thus, we only implement ACM-SGC.

- GCNII and GCNII*:

  GCNII: $\mathbf{H}^{(\ell+1)} = \sigma\left(\left(\left(1-\alpha_\ell\right)\hat{\mathbf{A}}\mathbf{H}^{(\ell)} + \alpha_\ell\mathbf{H}^{(0)}\right)\left(\left(1-\beta_\ell\right)\mathbf{I}_n + \beta_\ell\mathbf{W}^{(\ell)}\right)\right)$

  GCNII*: $\mathbf{H}^{(\ell+1)} = \sigma\left(\left(1-\alpha_\ell\right)\hat{\mathbf{A}}\mathbf{H}^{(\ell)}\left(\left(1-\beta_\ell\right)\mathbf{I}_n + \beta_\ell\mathbf{W}_1^{(\ell)}\right) + +\alpha_\ell\mathbf{H}^{(0)}\left(\left(1-\beta_\ell\right)\mathbf{I}_n + \beta_\ell\mathbf{W}_2^{(\ell)}\right)\right)$

  From the above architecture of GCNII and GCNII* we cam see that, without major modification, GCNII and GCNII* are hard to be put into ACMII framework. In ACMII framework, before apply $\hat{A}$, we first implement a nonlinear feature extractor $\sigma(H^\ell\mathbf{W}^{(\ell)})$. But in GCNII and GCNII*, before multiplying $W^\ell$(or $W_1^\ell, W_2^\ell$) to extract features, we need to add another term including $H^{(0)}$, which are not filtered by $\hat{A}$. This makes the order of aggregator $\hat{A}$ and nonlinear extractor unexchangable and thus, incompatible with ACMII framework. So we did not implement GCNII and GCNII* in ACMII framework.

The open source code is attached in the supplementary material.

## A.7 Computing Resources

For all experiments on synthetic datasets and real-world datasets, we use NVidia V100 GPUs with 16/32GB GPU memory, 8-core CPU, 16G Memory. The software implementation is based on PyTorch and PyTorch Geometric (Fey & Lenssen, 2019).

| Datasets | Models\Hyperparameters | lr | weight_decay | dropout | hidden | # layers | Gat heads | JK Type | lambda | alpha_l | results | std | average epoch time/average total time |
|---|---|---|---|---|---|---|---|---|---|---|---|---|---|
| | | | | | | **Hyperparameters for Baseline GNNs** | | | | | | | |
| **Cornell** | SGC-1 | 0.05 | 1.00E-02 | 0 | 64 | - | - | - | - | - | 70.98 | 8.39 | 2.53ms/0.51s |
| | SGC-2 | 0.05 | 1.00E-03 | 0 | 64 | - | - | - | - | - | 72.62 | 9.92 | 2.46ms/0.53s |
| | GCN | 0.1 | 5.00E-03 | 0.5 | 64 | 2 | - | - | - | - | 82.46 | 3.11 | 3.67ms/0.74s |
| | Snowball-2 | 0.01 | 5.00E-03 | 0.4 | 64 | 2 | - | - | - | - | 82.62 | 2.34 | 4.24ms/0.87s |
| | Snowball-3 | 0.01 | 5.00E-03 | 0.4 | 64 | 3 | - | - | - | - | 82.95 | 2.1 | 6.66ms/1.36s |
| | GCNII | 0.01 | 1.00E-03 | 0.5 | 64 | 16 | - | - | 0.5 | 0.5 | 89.18 | 3.96 | 25.41ms/8.11s |
| | GCNII* | 0.01 | 1.00E-03 | 0.5 | 64 | 8 | - | - | 0.5 | 0.5 | 90.49 | 4.45 | 15.35ms/4.05s |
| | FAGCN | 0.01 | 1.00E-04 | 0.7 | 32 | 2 | - | - | - | - | 88.03 | 5.6 | 8.1ms/3.8858s |
| | Mixhop | 0.01 | 0.001 | 0.5 | 16 | 2 | - | - | - | - | 60.33 | 28.53 | 10.379ms/2.105s |
| | H2GCN | 0.01 | 0.001 | 0.5 | 64 | 1 | - | - | - | - | 86.23 | 4.71 | 4.381ms/1.123s |
| | GCN+JK | 0.1 | 0.001 | 0.5 | 64 | 2 | - | cat | - | - | 66.56 | 13.82 | 5.589ms/1.227s |
| | GAT+JK | 0.1 | 0.001 | 0.5 | 32 | 2 | 8 | max | - | - | 74.43 | 10.24 | 10.725ms/2.478s |
| **Wisconsin** | SGC-1 | 0.05 | 5.00E-03 | 0 | 64 | - | - | - | - | - | 70.38 | 2.85 | 2.83ms/0.57s |
| | SGC-2 | 0.1 | 1.00E-03 | 0 | 64 | - | - | - | - | - | 74.75 | 2.89 | 2.14ms/0.43s |
| | GCN | 0.1 | 1.00E-03 | 0.7 | 64 | 2 | - | - | - | - | 75.5 | 2.92 | 3.74ms/0.76s |
| | Snowball-2 | 0.1 | 1.00E-03 | 0.5 | 64 | 2 | - | - | - | - | 74.88 | 3.42 | 3.73ms/0.76s |
| | Snowball-3 | 0.05 | 5.00E-04 | 0.8 | 64 | 3 | - | - | - | - | 69.5 | 5.01 | 5.46ms/1.12s |
| | GCNII | 0.01 | 1.00E-03 | 0.5 | 64 | 8 | - | - | 0.5 | 0.5 | 83.25 | 2.69 | |
| | GCNII* | 0.01 | 1.00E-03 | 0.5 | 64 | 4 | - | - | 1.5 | 0.3 | 89.12 | 3.06 | 9.26ms/1.96s |
| | FAGCN | 0.05 | 1.00E-04 | 0 | 32 | 2 | - | - | - | - | 89.75 | 6.37 | 12.9ms/4.6359s |
| | Mixhop | 0.01 | 0.001 | 0.5 | 16 | 2 | - | - | - | - | 77.25 | 7.80 | 10.281ms/2.095s |
| | H2GCN | 0.01 | 0.001 | 0.5 | 32 | 1 | - | - | - | - | 87.5 | 1.77 | 4.324ms/1.134s |
| | GCN+JK | 0.1 | 0.001 | 0.5 | 32 | 2 | - | cat | - | - | 62.5 | 15.75 | 5.117ms/1.049s |
| | GAT+JK | 0.1 | 0.001 | 0.5 | 4 | 2 | 8 | max | - | - | 69.5 | 3.12 | 10.762ms/2.25s |
| | APPNP | 0.05 | 0.001 | 0.5 | 64 | 2 | - | - | - | - | 92 | 3.59 | 10.303ms/2.104s |
| | GPRGNN | 0.05 | 0.001 | 0.5 | 256 | 2 | - | - | - | - | 93.75 | 2.37 | 11.856ms/2.415s |
| **Texas** | SGC-1 | 0.05 | 1.00E-03 | 0 | 64 | - | - | - | - | - | 83.28 | 5.43 | 2.55ms/0.54s |
| | SGC-2 | 0.01 | 1.00E-03 | 0 | 64 | - | - | - | - | - | 81.31 | 3.3 | 2.61ms/2.53s |
| | GCN | 0.05 | 1.00E-02 | 0.9 | 64 | 2 | - | - | - | - | 83.11 | 3.2 | 3.59ms/0.73s |
| | Snowball-2 | 0.05 | 1.00E-02 | 0.9 | 64 | 2 | - | - | - | - | 83.11 | 3.2 | 3.98ms/0.82s |
| | Snowball-3 | 0.05 | 1.00E-02 | 0.9 | 64 | 3 | - | - | - | - | 83.11 | 3.2 | 5.56ms/1.12s |
| | GCNII | 0.01 | 1.00E-04 | 0.5 | 64 | 4 | - | - | 1.5 | 0.5 | 82.46 | 4.58 | |
| | GCNII* | 0.01 | 1.00E-04 | 0.5 | 64 | 8 | - | - | 0.5 | 0.5 | 88.52 | 3.02 | 15.64ms/3.47s |
| | FAGCN | 0.01 | 5.00E-04 | 0 | 32 | 2 | - | - | - | - | 88.85 | 4.39 | 8.8ms/6.5252s |
| | Mixhop | 0.01 | 0.001 | 0.5 | 32 | 2 | - | - | - | - | 76.39 | 7.66 | 11.099ms/2.329s |
| | H2GCN | 0.01 | 0.001 | 0.5 | 64 | 1 | - | - | - | - | 85.90 | 3.53 | 4.197ms/0.95s |
| | GCN+JK | 0.1 | 0.001 | 0.5 | 32 | 2 | - | cat | - | - | 80.66 | 1.91 | 5.28ms/1.085s |
| | GAT+JK | 0.1 | 0.001 | 0.5 | 8 | 2 | 2 | cat | - | - | 75.41 | 7.18 | 10.937ms/2.402s |
| **Film** | SGC-1 | 0.01 | 5.00E-06 | 0 | 64 | - | - | - | - | - | 25.26 | 1.18 | 3.18ms/0.70s |
| | SGC-2 | 0.01 | 5.00E-06 | 0 | 64 | - | - | - | - | - | 28.81 | 1.11 | 2.13ms/0.43s |
| | GCN | 0.1 | 5.00E-04 | 0 | 64 | 2 | - | - | - | - | 35.51 | 0.99 | 4.86ms/0.99s |
| | Snowball-2 | 0.1 | 5.00E-04 | 0 | 64 | 2 | - | - | - | - | 35.97 | 0.66 | 5.59ms/1.14s |
| | Snowball-3 | 0.1 | 5.00E-04 | 0.2 | 64 | 3 | - | - | - | - | 36 | 1.36 | 7.89ms/1.60s |
| | GCNII | 0.01 | 1.00E-04 | 0.5 | 64 | 8 | - | - | 1.5 | 0.3 | 40.82 | 1.79 | 15.85ms/3.22s |
| | GCNII* | 0.01 | 1.00E-06 | 0.5 | 64 | 4 | - | - | 1 | 0.1 | 41.54 | 0.99 | |
| | FAGCN | 0.01 | 5.00E-05 | 0.6 | 32 | 2 | - | - | - | - | 31.59 | 1.37 | 45.4ms/11.107s |
| | Mixhop | 0.01 | 0.001 | 0.5 | 8 | 3 | 8 | max | - | - | 33.13 | 2.40 | 17.651ms/3.566s |
| | H2GCN | 0.01 | 0.001 | 0 | 64 | 1 | 8 | max | - | - | 38.85 | 1.17 | 8.101ms/1.695s |
| | GCN+JK | 0.01 | 0.001 | 0.5 | 64 | 2 | 8 | cat | - | - | 32.72 | 2.62 | 8.946ms/1.807s |
| | GAT+JK | 0.001 | 0.001 | 0.5 | 32 | 2 | 4 | cat | - | - | 35.41 | 0.97 | 20.726ms/4.187s |
| **Chameleon** | SGC-1 | 0.1 | 5.00E-06 | 0 | 64 | - | - | - | - | - | 64.86 | 1.81 | 3.48ms/2.96s |
| | SGC-2 | 0.1 | 0.00E+00 | 0 | 64 | - | - | - | - | - | 62.67 | 2.41 | 4.43ms/1.12s |
| | GCN | 0.01 | 1.00E-05 | 0.9 | 64 | 2 | - | - | - | - | 64.18 | 2.62 | 4.96ms/1.18s |
| | Snowball-2 | 1.00E-01 | 1.00E-05 | 0.9 | 64 | 2 | - | - | - | - | 64.99 | 2.39 | 4.96ms/1.00s |
| | Snowball-3 | 0.1 | 5.00E-06 | 0.9 | 64 | 3 | - | - | - | - | 65.49 | 1.64 | 7.44ms/1.50s |
| | GCNII | 0.01 | 5.00E-06 | 0.5 | 64 | 4 | - | - | 0.5 | 0.1 | 60.35 | 2.7 | 9.76ms/2.26s |
| | GCNII* | 0.01 | 5.00E-04 | 0.5 | 64 | 4 | - | - | 1.5 | 0.5 | 62.8 | 2.87 | 10.40ms/2.17s |
| | FAGCN | 0.002 | 1.00E-04 | 0 | 32 | 2 | - | - | - | - | 49.47 | 2.84 | 8.4ms/13.8696s |
| | Mixhop | 0.01 | 0.001 | 0.5 | 16 | 2 | 8 | max | - | - | 36.28 | 10.2 | 11.372ms/2.297s |
| | H2GCN | 0.01 | 0.001 | 0 | 32 | 1 | 8 | max | - | - | 52.3 | 0.48 | 4.059ms/0.82s |
| | GCN+JK | 0.001 | 0.001 | 0.5 | 32 | 2 | 8 | cat | - | - | 64.68 | 2.85 | 5.211ms/1.053s |
| | GAT+JK | 0.001 | 0.001 | 0.5 | 4 | 2 | 8 | max | - | - | 68.14 | 1.18 | 13.772ms/2.788s |
| **Squirrel** | SGC-1 | 0.05 | 0.00E+00 | 0 | 64 | - | - | - | - | - | 47.62 | 1.27 | 4.65ms/1.44s |
| | SGC-2 | 0.1 | 0.00E+00 | 0.9 | 64 | - | - | - | - | - | 41.25 | 1.4 | 35.06ms/7.81s |
| | GCN | 0.01 | 5.00E-05 | 0.7 | 64 | 2 | - | - | - | - | 44.76 | 1.39 | 8.41ms/2.50s |
| | Snowball-2 | 0.1 | 0.00E+00 | 0.9 | 64 | 2 | - | - | - | - | 47.88 | 1.23 | 8.96ms/1.92s |
| | Snowball-3 | 0.1 | 0.00E+00 | 0.8 | 64 | 3 | - | - | - | - | 48.25 | 0.94 | 14.00ms/2.90s |
| | GCNII | 0.01 | 1.00E-04 | 0.5 | 64 | 4 | - | - | 1.5 | 0.2 | 38.81 | 1.97 | 13.35ms/2.70s |
| | GCNII* | 0.01 | 5.00E-04 | 0.5 | 64 | 4 | - | - | 1.5 | 0.3 | 38.31 | 1.3 | 13.81ms/2.78s |
| | FAGCN | 0.05 | 1.00E-04 | 0 | 32 | 2 | - | - | - | - | 42.24 | 1.2 | 16ms/6.7961s |
| | Mixhop | 0.01 | 0.001 | 0.5 | 32 | 2 | - | - | - | - | 24.55 | 2.6 | 17.634ms/3.562s |
| | H2GCN | 0.01 | 0.001 | 0 | 16 | 1 | - | - | - | - | 30.39 | 1.22 | 9.315ms/1.882s |
| | GCN+JK | 0.001 | 0.001 | 0.5 | 32 | 2 | - | max | - | - | 53.4 | 1.9 | 14.321ms/2.905s |
| | GAT+JK | 0.001 | 0.001 | 0.5 | 8 | 2 | 4 | max | - | - | 52.28 | 3.61 | 29.097ms/5.878s |
| **Cora** | SGC-1 | 0.05 | 5.00E-06 | 0 | 64 | - | - | - | - | - | 85.12 | 1.64 | 3.47ms/11.55s |
| | SGC-2 | 0.1 | 1.00E-05 | 0 | 64 | - | - | - | - | - | 85.48 | 1.48 | 2.91ms/6.85s |
| | GCN | 0.1 | 5.00E-04 | 0.2 | 64 | 2 | - | - | - | - | 87.78 | 0.96 | 4.24ms/0.86s |
| | Snowball-2 | 0.1 | 5.00E-04 | 0.1 | 64 | 2 | - | - | - | - | 88.64 | 1.15 | 4.65ms/0.94s |
| | Snowball-3 | 0.05 | 1.00E-03 | 0.6 | 64 | 3 | - | - | - | - | 89.33 | 1.3 | 6.41ms/1.32s |
| | GCNII | 0.01 | 1.00E-04 | 0.5 | 64 | 16 | - | - | 0.5 | 0.2 | 88.98 | 1.33 | |
| | GCNII* | 0.01 | 5.00E-04 | 0.5 | 64 | 4 | - | - | 0.5 | 0.5 | 88.93 | 1.37 | 10.16ms/2.24s |
| | FAGCN | 0.05 | 5.00E-04 | 0 | 32 | 2 | - | - | - | - | 88.85 | 1.36 | 8.4ms/3.3183s |
| | Mixhop | 0.01 | 0.001 | 0.5 | 16 | 2 | - | - | - | - | 65.65 | 11.31 | 11.177ms/2.278s |
| | H2GCN | 0.01 | 0.001 | 0 | 32 | 1 | - | - | - | - | 87.52 | 0.61 | 4.335ms/1.209s |
| | GCN+JK | 0.001 | 0.001 | 0.5 | 64 | 2 | - | cat | - | - | 86.90 | 1.51 | 6.656ms/1.346s |
| | GAT+JK | 0.001 | 0.001 | 0.5 | 32 | 2 | 2 | cat | - | - | 89.52 | 0.43 | 12.91ms/2.608s |
| **CiteSeer** | SGC-1 | 0.1 | 5.00E-04 | 0 | 64 | - | - | - | - | - | 79.66 | 0.75 | 3.43ms/7.30s |
| | SGC-2 | 0.01 | 5.00E-04 | 0.9 | 64 | - | - | - | - | - | 80.75 | 1.15 | 5.33ms/4.40s |
| | GCN | 0.1 | 1.00E-03 | 0.9 | 64 | 2 | - | - | - | - | 81.39 | 1.23 | 4.18ms/0.86s |
| | Snowball-2 | 0.1 | 1.00E-03 | 0.8 | 64 | 2 | - | - | - | - | 81.53 | 1.71 | 5.19ms/1.11s |
| | Snowball-3 | 0.1 | 1.00E-03 | 0.9 | 64 | 3 | - | - | - | - | 80.93 | 1.32 | 7.64ms/1.69s |
| | GCNII | 0.01 | 1.00E-03 | 0.5 | 64 | 16 | - | - | 0.5 | 0.2 | 81.58 | 1.3 | |
| | GCNII* | 0.01 | 1.00E-03 | 0.5 | 64 | 16 | - | - | 0.5 | 0.2 | 81.83 | 1.78 | 32.50ms/10.29s |
| | FAGCN | 0.05 | 5.00E-04 | 0 | 32 | 2 | - | - | - | - | 82.37 | 1.46 | 9.4ms/4.7648s |
| | Mixhop | 0.01 | 0.001 | 0.5 | 16 | 2 | - | - | - | - | 49.52 | 13.35 | 13.793ms/2.786s |
| | H2GCN | 0.01 | 0.001 | 0 | 8 | 1 | - | - | - | - | 79.97 | 0.69 | 5.794ms/3.049s |
| | GCN+JK | 0.001 | 0.001 | 0.5 | 32 | 2 | - | max | - | - | 73.77 | 1.85 | 5.264ms/1.063s |
| | GAT+JK | 0.001 | 0.001 | 0.5 | 8 | 2 | 4 | max | - | - | 74.49 | 2.76 | 12.326ms/2.49s |
| **PubMed** | SGC-1 | 0.05 | 5.00E-06 | 0.3 | 64 | - | - | - | - | - | 87.75 | 0.88 | 6.04ms/2.61s |
| | SGC-2 | 0.05 | 5.00E-05 | 0.1 | 64 | - | - | - | - | - | 88.79 | 0.5 | 8.62ms/3.18s |
| | GCN | 0.1 | 5.00E-05 | 0.6 | 64 | 2 | - | - | - | - | 88.9 | 0.32 | 5.08ms/1.03s |
| | Snowball-2 | 0.1 | 5.00E-04 | 0 | 64 | 2 | - | - | - | - | 89.04 | 0.49 | 5.68ms/1.19s |
| | Snowball-3 | 0.1 | 5.00E-06 | 0 | 64 | 3 | - | - | - | - | 88.8 | 0.82 | 8.54ms/1.75s |
| | GCNII | 0.01 | 1.00E-06 | 0.5 | 64 | 4 | - | - | 0.5 | 0.5 | 89.8 | 0.3 | 10.98ms/3.21s |
| | GCNII* | 0.01 | 1.00E-06 | 0.5 | 64 | 4 | - | - | 0.5 | 0.1 | 89.98 | 0.52 | 11.47ms/3.24s |
| | FAGCN | 0.05 | 5.00E-04 | 0 | 32 | 2 | - | - | - | - | 89.98 | 0.54 | 14.5ms/6.411s |
| | Mixhop | 0.01 | 0.001 | 0.5 | 16 | 2 | - | - | - | - | 87.04 | 4.10 | 17.459ms/3.527s |
| | H2GCN | 0.01 | 0.001 | 0 | 64 | 1 | - | - | - | - | 87.78 | 0.28 | 8.039ms/2.28s |
| | GCN+JK | 0.01 | 0.001 | 0.5 | 32 | 2 | - | cat | - | - | 90.09 | 0.68 | 12.001ms/2.424s |
| | GAT+JK | 0.1 | 0.001 | 0.5 | 8 | 2 | 4 | max | - | - | 89.15 | 0.87 | 20.403ms/4.125s |
| **Deezer-Europe** | FAGCN | 0.01 | 0.0001 | 0 | 32 | 2 | - | - | - | - | 66.86 | 0.53 | 41.7ms/20.8362s |
| | GCNII | 0.01 | 5e-6,1e-5 | 0.5 | 64 | 32 | - | - | 0.5 | 0.5 | 66.38 | 0.45 | 126.58ms/63.16s |
| | GCNII* | 0.01 | 1e-4,1e-3 | 0.5 | 64 | 32 | - | - | 0.5 | 0.5 | 66.42 | 0.56 | 134.05ms/66.89s |

Table 6: Optimal Hyperparameters for baseline models

| Hyperparameters for ACM-GNNs and ACMII-GNNs | | | | | | | | | | | | | |
|---|---|---|---|---|---|---|---|---|---|---|---|---|---|
| Datasets | Models\Hyperparameters | lr | weight_decay | dropout | hidden | # layers | Gat heads | JK Type | lambda | alpha_l | results | std | average epoch time/average total time |
| **Cornell** | ACM-SGC-1 | 0.01 | 5.00E-03 | 0.6 | 64 | - | - | - | - | - | 93.77 | 1.91 | 5.53ms/2.31s |
| | ACM-SGC-2 | 0.01 | 5.00E-03 | 0.6 | 64 | - | - | - | - | - | 93.77 | 2.17 | 4.73ms/1.87s |
| | ACM-GCN | 0.05 | 1.00E-02 | 0.2 | 64 | 2 | - | - | - | - | 94.75 | 3.8 | 8.25ms/1.69s |
| | ACMII-GCN | 0.1 | 1.00E-02 | 0.5 | 64 | 2 | - | - | - | - | 95.25 | 2.79 | 8.43ms/1.71s |
| | ACM-GCNII | 0.01 | 1.00E-03 | 0.5 | 64 | 1 | - | - | 0.5 | 0.4 | 92.62 | 3.13 | 6.81ms/1.43s |
| | ACM-GCNII* | 0.01 | 5.00E-04 | 0.5 | 64 | 1 | - | - | 0.5 | 0.1 | 93.44 | 2.74 | 6.76ms/1.39s |
| | ACM-Snowball-2 | 0.05 | 1.00E-02 | 0.2 | 64 | 2 | - | - | - | - | 95.08 | 3.11 | 9.15ms/1.86s |
| | ACM-Snowball-3 | 0.1 | 1.00E-02 | 0.4 | 64 | 3 | - | - | - | - | 94.26 | 2.57 | 13.20ms/2.68s |
| | ACMII-Snowball-2 | 0.05 | 1.00E-02 | 0.6 | 64 | 2 | - | - | - | - | 95.25 | 1.55 | 8.23ms/1.72s |
| | ACMII-Snowball-3 | 0.05 | 1.00E-02 | 0.7 | 64 | 3 | - | - | - | - | 93.61 | 2.79 | 11.70ms/2.37s |
| **Wisconsin** | ACM-SGC-1 | 0.05 | 5.00E-03 | 0.7 | 64 | - | - | - | - | - | 93.25 | 2.92 | 5.96ms/1.34s |
| | ACM-SGC-2 | 0.1 | 5.00E-03 | 0.2 | 64 | - | - | - | - | - | 94 | 2.61 | 4.60ms/0.95s |
| | ACM-GCN | 0.1 | 5.00E-03 | 0 | 64 | 2 | - | - | - | - | 95.75 | 2.03 | 8.11ms/1.64s |
| | ACMII-GCN | 0.1 | 1.00E-02 | 0.2 | 64 | 2 | - | - | - | - | 96.62 | 2.44 | 8.28ms/1.68s |
| | ACM-GCNII | 0.01 | 5.00E-03 | 0.5 | 64 | 1 | - | - | 1 | 0.1 | 94.63 | 2.96 | 9.31ms/2.19s |
| | ACM-GCNII* | 0.01 | 1.00E-03 | 0.5 | 64 | 1 | - | - | 1.5 | 0.4 | 94.37 | 2.81 | 7.11ms/1.45s |
| | ACM-Snowball-2 | 0.1 | 5.00E-03 | 0.1 | 64 | 2 | - | - | - | - | 96.38 | 2.59 | 8.63ms/1.74s |
| | ACM-Snowball-3 | 0.05 | 1.00E-02 | 0.3 | 64 | 3 | - | - | - | - | 96.62 | 1.86 | 12.79ms/2.58s |
| | ACMII-Snowball-2 | 0.1 | 1.00E-02 | 0.1 | 64 | 2 | - | - | - | - | 96.63 | 2.24 | 8.11ms/1.65s |
| | ACMII-Snowball-3 | 0.1 | 5.00E-03 | 0.1 | 64 | 3 | - | - | - | - | 97 | 2.63 | 12.38ms/2.51s |
| **Texas** | ACM-SGC-1 | 0.01 | 5.00E-03 | 0.6 | 64 | - | - | - | - | - | 93.61 | 1.55 | 5.43ms/2.18s |
| | ACM-SGC-2 | 0.05 | 5.00E-03 | 0.4 | 64 | - | - | - | - | - | 93.44 | 2.54 | 4.59ms/1.01s |
| | ACM-GCN | 0.05 | 1.00E-02 | 0.6 | 64 | 2 | - | - | - | - | 94.92 | 2.88 | 8.33ms/1.70s |
| | ACMII-GCN | 0.1 | 5.00E-03 | 0.4 | 64 | 2 | - | - | - | - | 95.08 | 2.54 | 8.49ms/1.72s |
| | ACM-GCNII | 0.01 | 1.00E-03 | 0.5 | 64 | 1 | - | - | 0.5 | 0.4 | 92.46 | 1.97 | 6.47ms/1.36s |
| | ACM-GCNII* | 0.01 | 1.00E-03 | 0.5 | 64 | 1 | - | - | 0.5 | 0.4 | 93.28 | 2.79 | 7.03ms/1.45s |
| | ACM-Snowball-2 | 0.05 | 1.00E-02 | 0.1 | 64 | 2 | - | - | - | - | 95.74 | 2.22 | 8.35ms/1.71s |
| | ACM-Snowball-3 | 0.01 | 5.00E-03 | 0.6 | 64 | 3 | - | - | - | - | 94.75 | 2.41 | 12.56ms/2.63s |
| | ACMII-Snowball-2 | 0.1 | 1.00E-02 | 0.4 | 64 | 2 | - | - | - | - | 95.25 | 1.55 | 9.74ms/1.97s |
| | ACMII-Snowball-3 | 0.05 | 1.00E-02 | 0.6 | 64 | 3 | - | - | - | - | 94.75 | 3.09 | 11.91ms/2.42s |
| **Film** | ACM-SGC-1 | 0.05 | 5.00E-05 | 0.7 | 64 | - | - | - | - | - | 39.33 | 1.25 | 5.21ms/2.33s |
| | ACM-SGC-2 | 0.1 | 5.00E-05 | 0.7 | 64 | - | - | - | - | - | 40.13 | 1.21 | 12.41ms/4.87s |
| | ACM-GCN | 0.1 | 5.00E-04 | 0.5 | 64 | 2 | - | - | - | - | 41.62 | 1.15 | 10.72ms/2.66s |
| | ACMII-GCN | 0.1 | 5.00E-04 | 0.5 | 64 | 2 | - | - | - | - | 41.24 | 1.16 | 10.51ms/2.44s |
| | ACM-GCNII | 0.01 | 0.00E+00 | 0.5 | 64 | 3 | - | - | 1.5 | 0.2 | 41.37 | 1.37 | 13.65ms/2.74s |
| | ACM-GCNII* | 0.01 | 1.00E-05 | 0.5 | 64 | 3 | - | - | 1.5 | 0.1 | 41.27 | 1.24 | 14.98ms/3.01s |
| | ACM-Snowball-2 | 0.1 | 5.00E-03 | 0 | 64 | 2 | - | - | - | - | 41.4 | 1.23 | 10.30ms/2.08s |
| | ACM-Snowball-3 | 0.05 | 1.00E-02 | 0 | 64 | 3 | - | - | - | - | 41.27 | 0.8 | 16.43ms/3.52s |
| | ACMII-Snowball-2 | 0.1 | 5.00E-03 | 0 | 64 | 2 | - | - | - | - | 41.1 | 0.75 | 10.74ms/2.19s |
| | ACMII-Snowball-3 | 0.05 | 1.00E-04 | 0.2 | 64 | 3 | - | - | - | - | 40.31 | 1.6 | 16.31ms/3.29s |
| **Chameleon** | ACM-SGC-1 | 0.1 | 5.00E-06 | 0.9 | 64 | - | - | - | - | - | 63.68 | 1.62 | 5.41ms/1.21s |
| | ACM-SGC-2 | 0.1 | 5.00E-06 | 0.9 | 64 | - | - | - | - | - | 60.48 | 1.55 | 7.86ms/1.81s |
| | ACM-GCN | 0.01 | 5.00E-06 | 0.8 | 64 | 2 | - | - | - | - | 68.18 | 1.67 | 10.55ms/3.12s |
| | ACMII-GCN | 0.05 | 5.00E-05 | 0.7 | 64 | 2 | - | - | - | - | 68.38 | 1.36 | 10.90ms/2.39s |
| | ACM-GCNII | 0.01 | 5.00E-06 | 0.5 | 64 | 4 | - | - | 0.5 | 0.1 | 58.73 | 2.52 | 18.31ms/3.68s |
| | ACM-GCNII* | 0.01 | 1.00E-03 | 0.5 | 64 | 1 | - | - | 1 | 0.1 | 61.66 | 2.29 | 6.68ms/1.40s |
| | ACM-Snowball-2 | 0.05 | 5.00E-05 | 0.7 | 64 | 2 | - | - | - | - | 68.51 | 1.7 | 9.92ms/2.06s |
| | ACM-Snowball-3 | 0.01 | 1.00E-04 | 0.7 | 64 | 3 | - | - | - | - | 68.4 | 2.05 | 14.49ms/3.15s |
| | ACMII-Snowball-2 | 0.1 | 5.00E-05 | 0.6 | 64 | 2 | - | - | - | - | 67.83 | 2.63 | 9.99ms/2.10s |
| | ACMII-Snowball-3 | 0.05 | 1.00E-04 | 0.7 | 64 | 3 | - | - | - | - | 67.53 | 2.83 | 15.03ms/3.29s |
| **Squirrel** | ACM-SGC-1 | 0.05 | 0.00E+00 | 0.9 | 64 | - | - | - | - | - | 46.4 | 1.13 | 6.96ms/2.16s |
| | ACM-SGC-2 | 0.05 | 0.00E+00 | 0.9 | 64 | - | - | - | - | - | 40.91 | 1.39 | 35.20ms/10.66s |
| | ACM-GCN | 0.05 | 5.00E-06 | 0.6 | 64 | 2 | - | - | - | - | 58.02 | 1.86 | 14.35ms/2.98s |
| | ACMII-GCN | 0.05 | 0.00E+00 | 0.7 | 64 | 2 | - | - | - | - | 53.76 | 1.63 | 14.08ms/3.39s |
| | ACM-GCNII | 0.01 | 1.00E-05 | 0.5 | 64 | 4 | - | - | 0.5 | 0.1 | 40.9 | 1.58 | 20.72ms/4.17s |
| | ACM-GCNII* | 0.01 | 5.00E-06 | 0.5 | 64 | 4 | - | - | 0.5 | 0.3 | 38.32 | 1.5 | 21.78ms/4.38s |
| | ACM-Snowball-2 | 0.05 | 5.00E-06 | 0.6 | 64 | 2 | - | - | - | - | 55.97 | 2.03 | 15.38ms/3.15s |
| | ACM-Snowball-3 | 0.01 | 1.00E-04 | 0.6 | 64 | 3 | - | - | - | - | 55.73 | 2.39 | 26.15ms/5.94s |
| | ACMII-Snowball-2 | 0.1 | 5.00E-06 | 0.6 | 64 | 2 | - | - | - | - | 53.48 | 0.6 | 15.54ms/3.19s |
| | ACMII-Snowball-3 | 0.05 | 5.00E-05 | 0.5 | 64 | 3 | - | - | - | - | 52.31 | 1.57 | 26.24ms/5.30s |
| **Cora** | ACM-SGC-1 | 0.01 | 5.00E-06 | 0.9 | 64 | - | - | - | - | - | 86.63 | 1.13 | 6.00ms/7.40s |
| | ACM-SGC-2 | 0.1 | 5.00E-05 | 0.6 | 64 | - | - | - | - | - | 87.64 | 0.99 | 4.85ms/1.17s |
| | ACM-GCN | 0.1 | 5.00E-03 | 0.5 | 64 | 2 | - | - | - | - | 88.62 | 1.22 | 8.84ms/1.81s |
| | ACMII-GCN | 0.1 | 5.00E-04 | 0.4 | 64 | 2 | - | - | - | - | 89 | 0.72 | 8.93ms/1.83s |
| | ACM-GCNII | 0.01 | 1.00E-03 | 0.5 | 64 | 3 | - | - | 1 | 0.2 | 89.1 | 1.61 | 14.07ms/3.04s |
| | ACM-GCNII* | 0.01 | 1.00E-02 | 0.5 | 64 | 4 | - | - | 1 | 0.2 | 89 | 1.35 | 11.36ms/2.48s |
| | ACM-Snowball-2 | 0.05 | 1.00E-03 | 0.6 | 64 | 2 | - | - | - | - | 88.83 | 1.49 | 9.34ms/1.92s |
| | ACM-Snowball-3 | 0.1 | 1.00E-02 | 0.3 | 64 | 3 | - | - | - | - | 89.59 | 1.58 | 13.33ms/2.75s |
| | ACMII-Snowball-2 | 0.1 | 5.00E-03 | 0.5 | 64 | 2 | - | - | - | - | 88.95 | 1.04 | 9.29ms/1.90s |
| | ACMII-Snowball-3 | 0.1 | 5.00E-03 | 0.5 | 64 | 3 | - | - | - | - | 89.36 | 1.26 | 14.18ms/2.89s |
| **CiteSeer** | ACM-SGC-1 | 0.01 | 5.00E-04 | 0.9 | 64 | - | - | - | - | - | 80.96 | 0.93 | 5.90ms/4.31s |
| | ACM-SGC-2 | 0.05 | 5.00E-04 | 0.9 | 64 | - | - | - | - | - | 80.93 | 1.16 | 5.01ms/1.42s |
| | ACM-GCN | 0.05 | 5.00E-03 | 0.7 | 64 | 2 | - | - | - | - | 81.68 | 0.97 | 11.35ms/2.57s |
| | ACMII-GCN | 0.05 | 5.00E-05 | 0.7 | 64 | 2 | - | - | - | - | 81.58 | 1.77 | 9.55ms/1.94s |
| | ACM-GCNII | 0.01 | 1.00E-02 | 0.5 | 64 | 3 | - | - | 0.5 | 0.3 | 82.28 | 1.12 | 15.61ms/3.56s |
| | ACM-GCNII* | 0.01 | 1.00E-02 | 0.5 | 64 | 3 | - | - | 0.5 | 0.5 | 81.69 | 1.25 | 15.56ms/3.61s |
| | ACM-Snowball-2 | 0.05 | 5.00E-03 | 0.7 | 64 | 2 | - | - | - | - | 81.58 | 1.23 | 11.14ms/2.50s |
| | ACM-Snowball-3 | 0.01 | 5.00E-03 | 0.9 | 64 | 3 | - | - | - | - | 81.32 | 0.97 | 15.91ms/5.36s |
| | ACMII-Snowball-2 | 0.05 | 5.00E-03 | 0.7 | 64 | 2 | - | - | - | - | 82.07 | 1.04 | 10.97ms/2.55s |
| | ACMII-Snowball-3 | 0.05 | 1.00E-04 | 0.6 | 64 | 3 | - | - | - | - | 81.56 | 1.15 | 14.95ms/3.03s |
| **PubMed** | ACM-SGC-1 | 0.05 | 5.00E-06 | 0.3 | 64 | - | - | - | - | - | 87.75 | 0.88 | 6.04ms/2.61s |
| | ACM-SGC-2 | 0.05 | 5.00E-05 | 0.1 | 64 | - | - | - | - | - | 88.79 | 0.5 | 8.62ms/3.18s |
| | ACM-GCN | 0.1 | 5.00E-04 | 0.2 | 64 | 2 | - | - | - | - | 90.54 | 0.63 | 10.20ms/2.08s |
| | ACMII-GCN | 0.1 | 5.00E-04 | 0.2 | 64 | 2 | - | - | - | - | 90.74 | 0.5 | 10.20ms/2.07s |
| | ACM-GCNII | 0.01 | 1.00E-04 | 0.5 | 64 | 3 | - | - | 1.5 | 0.5 | 90.12 | 0.4 | 15.07ms/3.35s |
| | ACM-GCNII* | 0.01 | 1.00E-04 | 0.5 | 64 | 3 | - | - | 1.5 | 0.5 | 90.18 | 0.51 | 16.62ms/3.72s |
| | ACM-Snowball-2 | 0.1 | 1.00E-04 | 0.3 | 64 | 2 | - | - | - | - | 90.81 | 0.52 | 11.52ms/2.36s |
| | ACM-Snowball-3 | 0.05 | 1.00E-03 | 0.2 | 64 | 3 | - | - | - | - | 91.44 | 0.59 | 18.06ms/3.69s |
| | ACMII-Snowball-2 | 0.1 | 1.00E-04 | 0.3 | 64 | 2 | - | - | - | - | 90.56 | 0.39 | 11.74ms/2.39s |
| | ACMII-Snowball-3 | 0.1 | 5.00E-04 | 0.2 | 64 | 3 | - | - | - | - | 91.31 | 0.6 | 18.61ms/3.88s |
| **Deezer-Europe** | ACM-SGC-1 | 0.05 | 0,5e-6,1e-5,5e-5 | 0.3 | 64 | - | - | - | - | - | 66.67 | 0.56 | 146.41ms/73.06s |
| | ACM-SGC-2 | 0.002 | 5e-5,1e-4 | 0.3 | 64 | - | - | - | - | - | 66.53 | 0.57 | 195.21ms/97.41s |
| | ACM-GCN | 0.002 | 5.00E-04 | 0.5 | 64 | 2 | - | - | - | - | 67.01 | 0.38 | 136.45ms/68.09s |
| | ACMII-GCN | 0.01 | 5.00E-05 | 0.8 | 64 | 2 | - | - | - | - | 67.15 | 0.41 | 135.24ms/67.48s |
| | ACM-GCNII | 0.01 | 0,5e-6 | 0.5 | 64 | 1 | - | - | 0.5 | 0.4 | 66.39 | 0.56 | 80.82ms/40.33s |
| | ACM-GCNII* | 0.01 | 0.0001,1e-3 | 0.5 | 64 | 1 | - | - | 1.5 | 0.2 | 66.6 | 0.57 | 80.95ms/40.40s |

Table 7: Optimal Hyperparameters for ACM-GNNs and ACMII-GNNs

## B  EXPERIMENTAL SETUP AND FURTHER DISCUSSION ON SYNTHETIC GRAPHS

### B.1  DETAILED DESCRIPTION OF DATA GENERATION PROCESS

- For each node $v$, we first randomly generate its degree $d_v$.
- Given $d_v$, we sample $hd_v$ intra-class edges and $(1 - h)d_v$ inter-class edges.

More specifically in our synthetic experiments, for a given $h$,

- we generate node degree $d_v$ for nodes in each class from multinomial distribution with
  `numpy.random.multinomial(800/h, numpy.ones(400)/400, size=1)[0]`.
- For a sampled $d_v$, we generate intra-class edges from (does not include self-loop)
  `numpy.random.multinomial(hd_v, numpy.ones(399)/399, size=1)[0]`
  and inter-class edges from `numpy.random.multinomial((1-h) d_v, numpy.ones(1600)/1600, size=1)[0]`.

We will release the code and the generated data later.

### B.2  FURTHER DISCUSSION OF AGGREGATION HOMOPHILY ON REGULAR GRAPHS

We notice that in Figure 2(a), the performance of SGC-1 and GCN both have a turning point, *i.e.,* when $H_{\text{edge}}(\mathcal{G})$ is smaller than a certain value, the performance will get better instead of getting worse. With some extra restriction on node degree in data generation process, we find that this interesting phenomenon can be theoretically explained by the following proposition 1 based on our proposed similarity matrix which can verify the usefulness of $H_{\text{agg}}^M(\mathcal{G})$. We first generate regular graphs ,*i.e.,* each node has the same degree, as follows,

**Generate Synthetic Regular Graphs**   We first generate 180 graphs in total with 18 edge homophily levels varied from 0.05 to 0.9, each corresponding to 10 graphs. For every generated graph, we have 5 classes with 400 nodes in each class. For each node, we randomly generate 10 intra-class edges and $[\frac{10}{H_{\text{edge}}(\mathcal{G})} - 10]$ inter-class edges. The features of nodes in each class are sampled from node features in the corresponding class of the base dataset. Nodes are randomly split into 60%/20%/20% for train/validation/test. We train 1-hop SGC (*sgc-1*) Wu et al. (2019) and GCN Kipf & Welling (2016) on synthetic data (see Appendix A.1 for hyperparameter searching range). For each value of $H_{\text{edge}}(\mathcal{G})$, we take the average test accuracy and standard deviation of runs over 10 generated graphs. We plot the performance curves in Figure 5.

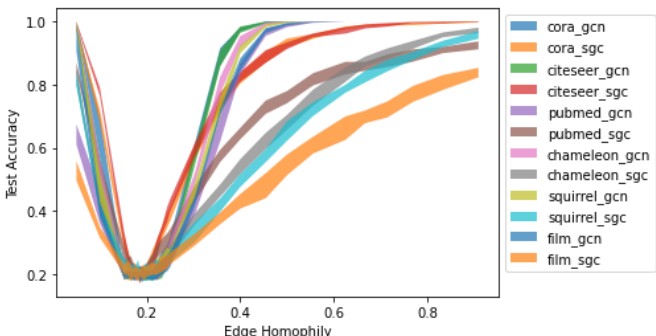

Figure 5: Synthetic experiments for edge homophily on regular graphs.

From Figure 5 we can see that the turning point is a bit less than 0.2. We derive the following proposition for $d$-regular graph to explain and predict it.

**Proposition 1.** (See Appendix D for proof). Suppose there are $C$ classes in the graph $\mathcal{G}$ and $\mathcal{G}$ is a $d$-regular graph (each node has $d$ neighbors). Given $d$, edges for each node are *i.i.d.* generated, such

that each edge of any node has probability $h$ to connect with nodes in the same class and probability $1 - h$ to connect with nodes in different classes. Let the aggregation operator $\hat{A} = \hat{A}_{\text{rw}}$. Then, for nodes $v$, $u_1$ and $u_2$, where $Z_{u_1,:} = Z_{v,:}$ and $Z_{u_2,:} \neq Z_{v,:}$, we have

$$g(h) \equiv \mathbb{E}\left(S(\hat{A}, Z)_{v,u_1}\right) - \mathbb{E}\left(S(\hat{A}, Z)_{v,u_2}\right) = \left(\frac{(C-1)(hd+1) - (1-h)d}{(C-1)(d+1)}\right)^2 \tag{12}$$

and the minimum of $g(h)$ is reached at

$$h = \frac{d + 1 - C}{Cd} = \frac{d_{\text{intra}}/h + 1 - C}{C(d_{\text{intra}}/h)} \Rightarrow h = \frac{d_{\text{intra}}}{Cd_{\text{intra}} + C - 1}$$

where $d_{\text{intra}} = dh$, which is the expected number of neighbors of a node that have the same label as the node.

The value of $g(h)$ in equation 12 is the expected differences of the similarity values between nodes in the same class as $v$ and nodes in other classes. $g(h)$ is strongly related to the definition of aggregation homophily and its minimum potentially implies the turning point of performance curves. In the synthetic experiments, we have $d_{\text{intra}} = 10, C = 5$ and the minimum of $g(h)$ is reached at $h = 5/27 \approx 0.1852$, which corresponds to the lowest point in the performance curve in Figure 5. In other words, the $H_{\text{edge}}(\mathcal{G})$ where SGC-1 and GCN perform worst is where $g(h)$ gets the smallest value, instead of the point with the smallest edge homophily value $H_{\text{edge}}(\mathcal{G}) = 0$. This reveals the advantage of $H_{\text{agg}}(\mathcal{G})$ over $H_{\text{edge}}(\mathcal{G})$ by taking use of the similarity matrix.

## C  DETAILS OF GRADIENT CALCULATION IN EQUATION 5

### C.1  DERIVATION IN MATRIX FORM

This derivation is similar to (Luan et al., 2020a).

In output layer, we have

$$Y = \text{softmax}(\hat{A}XW) \equiv \text{softmax}(Y') = \left(\exp(Y')\mathbf{1}_C\mathbf{1}_C^T\right)^{-1} \odot \exp(Y') > 0$$
$$\mathcal{L} = -\text{trace}(Z^T \log Y)$$

where $\mathbf{1}_C \in \mathcal{R}^{C \times 1}$, $(\cdot)^{-1}$ is point-wise inverse function and each element of $Y$ is positive. Then

$$d\mathcal{L} = -\text{trace}\left(Z^T((Y)^{-1} \odot dY)\right) = -\text{trace}\left(Z^T\left((\text{softmax}(Y'))^{-1} \odot d\,\text{softmax}(Y')\right)\right)$$

Note that

$$
\begin{aligned}
d\,\text{softmax}(Y') = &-\left(\exp(Y')\mathbf{1}_C\mathbf{1}_C^T\right)^{-2} \odot [(\exp(Y') \odot dY')\mathbf{1}_C\mathbf{1}_C^T] \odot \exp(Y') \\
&+ \left(\exp(Y')\mathbf{1}_C\mathbf{1}_C^T\right)^{-1} \odot (\exp(Y') \odot dY') \\
= &-\text{softmax}(Y') \odot \left(\exp(Y')\mathbf{1}_C\mathbf{1}_C^T\right)^{-1} \odot [(\exp(Y') \odot dY')\mathbf{1}_C\mathbf{1}_C^T] \\
&+ \text{softmax}(Y') \odot dY' \\
= &\text{softmax}(Y') \odot \left(-\left(\exp(Y')\mathbf{1}_C\mathbf{1}_C^T\right)^{-1} \odot [(\exp(Y') \odot dY')\mathbf{1}_C\mathbf{1}_C^T] + dY'\right)
\end{aligned}
$$

Then,

$$
\begin{aligned}
d\mathcal{L} = & -\operatorname{trace}\Bigg( Z^T \Bigg( (\operatorname{softmax}(Y'))^{-1} \odot \Big[ \operatorname{softmax}(Y') \odot \Big( -\left(\exp(Y')\mathbf{1}_C\mathbf{1}_C^T\right)^{-1} \\
& \odot \left[ (\exp(Y') \odot dY')\mathbf{1}_C\mathbf{1}_C^T \right] + dY' \Big) \Big] \Bigg) \Bigg) \\
= & -\operatorname{trace}\Big( Z^T \Big( -\left(\exp(Y')\mathbf{1}_C\mathbf{1}_C^T\right)^{-1} \odot \left[ (\exp(Y') \odot dY')\mathbf{1}_C\mathbf{1}_C^T \right] + dY' \Big) \Big) \\
= & \operatorname{trace}\Bigg( \Big( \Big( Z \odot \left(\exp(Y')\mathbf{1}_C\mathbf{1}_C^T\right)^{-1} \Big) \mathbf{1}_C\mathbf{1}_C^T \Big)^T [\exp(Y') \odot dY'] - Z^T dY' \Bigg) \\
= & \operatorname{trace}\Bigg( \Big( \exp(Y') \odot \Big( \Big( Z \odot \left(\exp(Y')\mathbf{1}_C\mathbf{1}_C^T\right)^{-1} \Big) \mathbf{1}_C\mathbf{1}_C^T \Big) \Big)^T dY' - Z^T dY' \Bigg) \\
= & \operatorname{trace}\Bigg( \Big( \exp(Y') \odot \left(\exp(Y')\mathbf{1}_C\mathbf{1}_C^T\right)^{-1} \Big)^T dY' - Z^T dY' \Bigg) \\
= & \operatorname{trace}\Big( (\operatorname{softmax}(Y') - Z)^T dY' \Big)
\end{aligned}
$$

where the 4-th equation holds due to $\Big( Z \odot \left(\exp(Y')\mathbf{1}_C\mathbf{1}_C^T\right)^{-1} \Big) \mathbf{1}_C\mathbf{1}_C^T = \left(\exp(Y')\mathbf{1}_C\mathbf{1}_C^T\right)^{-1}$. Thus, we have

$$
\frac{d\mathcal{L}}{dY'} = \operatorname{softmax}(Y') - Z = Y - Z
$$

For $Y'$ and $W$, we have

$$
dY' = \hat{A} X dW \text{ and } d\mathcal{L} = \operatorname{trace}\left( \frac{d\mathcal{L}}{dY'}^T dY' \right) = \operatorname{trace}\left( \frac{d\mathcal{L}}{dY'}^T \hat{A} X \, dW \right) = \operatorname{trace}\left( \frac{d\mathcal{L}}{dW}^T dW \right)
$$

To get $\frac{d\mathcal{L}}{dW}$ we have,

$$
\frac{d\mathcal{L}}{dW} = X^T \hat{A}^T \frac{d\mathcal{L}}{dY'} = X^T \hat{A}^T (Y - Z) \tag{13}
$$

## C.2 COMPONENT-WISE DERIVATION

Denote $\tilde{X} = XW$. We rewrite $\mathcal{L}$ as follows:

$$
\begin{aligned}
\mathcal{L} = & -\operatorname{trace}\left( Z^T \log \left( (\exp(Y')\mathbf{1}_C\mathbf{1}_C^T)^{-1} \odot \exp(Y') \right) \right) \\
= & -\operatorname{trace}\left( Z^T \left( -\log(\exp(Y')\mathbf{1}_C\mathbf{1}_C^T) + Y' \right) \right) \\
= & -\operatorname{trace}\left( Z^T Y' \right) + \operatorname{trace}\left( Z^T \log \left( \exp(Y')\mathbf{1}_C\mathbf{1}_C^T \right) \right) \\
= & -\operatorname{trace}\left( Z^T \hat{A} X W \right) + \operatorname{trace}\left( Z^T \log \left( \exp(Y')\mathbf{1}_C\mathbf{1}_C^T \right) \right) \\
= & -\operatorname{trace}\left( Z^T \hat{A} X W \right) + \operatorname{trace}\left( \mathbf{1}_C^T \log \left( \exp(Y')\mathbf{1}_C \right) \right) \\
= & -\sum_{i=1}^N \sum_{j \in \mathcal{N}_i} \hat{A}_{i,j} Z_{i,:} \tilde{X}_{j,:}^T + \sum_{i=1}^N \log \left( \sum_{c=1}^C \exp( \sum_{j \in \mathcal{N}_i} \hat{A}_{i,j} \tilde{X}_{j,c}) \right) \\
= & -\sum_{i=1}^N \log \left( \exp \left( \sum_{c=1}^C \sum_{j \in \mathcal{N}_i} \hat{A}_{i,j} Z_{i,c} \tilde{X}_{j,c} \right) \right) + \sum_{i=1}^N \log \left( \sum_{c=1}^C \exp \left( \sum_{j \in \mathcal{N}_i} \hat{A}_{i,j} \tilde{X}_{j,c} \right) \right) \\
= & -\sum_{i=1}^N \log \frac{\exp \left( \sum_{c=1}^C \sum_{j \in \mathcal{N}_i} \hat{A}_{i,j} Z_{i,c} \tilde{X}_{j,c} \right)}{\left( \sum_{c=1}^C \exp( \sum_{j \in \mathcal{N}_i} \hat{A}_{i,j} \tilde{X}_{j,c}) \right)}
\end{aligned}
$$

Note that $\sum\limits_{c=1}^{C} Z_{j,c} = 1$ for any $j$. Consider the derivation of $\mathcal{L}$ over $\tilde{X}_{j',c'}$:

$$\frac{d\mathcal{L}}{d\tilde{X}_{j',c'}}$$

$$= -\sum_{i=1}^{N} \frac{\sum\limits_{c=1}^{C} \exp(\sum\limits_{j\in\mathcal{N}_i} \hat{A}_{i,j}\tilde{X}_{j,c})}{\exp\left(\sum\limits_{c=1}^{C}\sum\limits_{j\in\mathcal{N}_i} \hat{A}_{i,j}Z_{i,c}\tilde{X}_{j,c}\right)}$$

$$\times \left( \frac{\left(\hat{A}_{i,j'}Z_{i,c'}\right)\exp\left(\sum\limits_{c=1}^{C}\sum\limits_{j\in\mathcal{N}_i} \hat{A}_{i,j}Z_{i,c}\tilde{X}_{j,c}\right)\left(\sum\limits_{c=1}^{C}\exp(\sum\limits_{j\in\mathcal{N}_i}\hat{A}_{i,j}\tilde{X}_{j,c})\right)}{\left(\sum\limits_{c=1}^{C}\exp(\sum\limits_{j\in\mathcal{N}_i}\hat{A}_{i,j}\tilde{X}_{j,c})\right)^2} \right.$$

$$\left. - \frac{\left(\hat{A}_{i,j'}\right)\exp\left(\sum\limits_{c=1}^{C}\sum\limits_{j\in\mathcal{N}_i} \hat{A}_{i,j}Z_{i,c}\tilde{X}_{j,c}\right)\left(\exp(\sum\limits_{j\in\mathcal{N}_i}\hat{A}_{i,j}\tilde{X}_{j,c'})\right)}{\left(\sum\limits_{c=1}^{C}\exp(\sum\limits_{j\in\mathcal{N}_i}\hat{A}_{i,j}\tilde{X}_{j,c})\right)^2} \right)$$

$$= -\sum_{i=1}^{N}\left( \frac{\left(\hat{A}_{i,j'}Z_{i,c'}\right)\left(\sum\limits_{c=1}^{C}\exp(\sum\limits_{j\in\mathcal{N}_i}\hat{A}_{i,j}\tilde{X}_{j,c})\right) - \left(\hat{A}_{i,j'}\right)\left(\exp(\sum\limits_{j\in\mathcal{N}_i}\hat{A}_{i,j}\tilde{X}_{j,c'})\right)}{\left(\sum\limits_{c=1}^{C}\exp(\sum\limits_{j\in\mathcal{N}_i}\hat{A}_{i,j}\tilde{X}_{j,c})\right)} \right)$$

$$= -\sum_{i=1}^{N}\left( \hat{A}_{i,j'}\frac{\left(\sum\limits_{c=1,c\neq c'}^{C}(Z_{i,c'})\exp(\sum\limits_{j\in\mathcal{N}_i}\hat{A}_{i,j}\tilde{X}_{j,c})\right) + (Z_{i,c'}-1)\left(\exp(\sum\limits_{j\in\mathcal{N}_i}\hat{A}_{i,j}\tilde{X}_{j,c'})\right)}{\left(\sum\limits_{c=1}^{C}\exp(\sum\limits_{j\in\mathcal{N}_i}\hat{A}_{i,j}\tilde{X}_{j,c})\right)} \right)$$

$$= -\sum_{i=1}^{N}\hat{A}_{i,j'}\left(Z_{i,c'}\hat{P}(Y_i\neq c') + (Z_{i,c'}-1)\hat{P}(Y_i = c')\right)$$

$$= -\sum_{i=1}^{N}\hat{A}_{i,j'}\left(Z_{i,c'} - \hat{P}(Y_i = c')\right)$$

Writing the above in matrix form, we have

$$\frac{d\mathcal{L}}{d\tilde{X}} = \hat{A}(Z-Y),\ \frac{d\mathcal{L}}{d\tilde{W}} = X^T\hat{A}^T(Z-Y),\ \Delta Y' \propto \hat{A}XX^T\hat{A}^T(Z-Y) \tag{14}$$

## D  PROOF OF PROPOSITION 1

*Proof.* According to the given assumptions, for node $v$, we have $\hat{A}_{v,k} = \frac{1}{d+1}$, the expected number of intra-class edges is $dh$ (here the self-loop edge introduced by $\hat{A}$ is not counted based on the definition of edge homophily and data generation process) and inter-class edges is $(1-h)d$. Suppose there are $C\geq 2$ classes. Consider matrix $\hat{A}Z$,

Then, we have $\mathbb{E}\left[(\hat{A}Z)_{v,c}\right] = \mathbb{E}\left[\sum\limits_{k\in\mathcal{V}}\hat{A}_{v,k}\mathbf{1}_{\{Z_{k,:}=e_c^T\}}\right] = \sum\limits_{k\in\mathcal{V}}\frac{\mathbb{E}\left[\mathbf{1}_{\{Z_{k,:}=e_c^T\}}\right]}{d+1}$, where $\mathbf{1}$ is the indicator function.

When $v$ is in class $c$, we have $\sum_{k \in \mathcal{V}} \frac{\mathbb{E}\left[\mathbf{1}_{\{Z_{k,:}=e_c^T\}}\right]}{d+1} = \frac{hd+1}{d+1}$ ($hd+1 = hd$ intra-class edges $+\,1$ self-loop introduced by $\hat{A}$).

When $v$ is not in class $c$, we have $\sum_{k \in \mathcal{V}} \frac{\mathbb{E}\left[\mathbf{1}_{\{Z_{k,:}=e_c^T\}}\right]}{d+1} = \frac{(1-h)d}{(C-1)(d+1)}$ ($(1-h)d$ inter-class edges uniformly distributed in the other $C-1$ classes).

For nodes $v, u$, we have $(\hat{A}Z)_{v,:}, (\hat{A}Z)_{u,:} \in \mathbb{R}^C$ and since elements in $\hat{A}_{v,k}$ and $\hat{A}_{u,k'}$ are independently generated for all $k, k' \in \mathcal{V}$, we have

$$\mathbb{E}\left[(\hat{A}Z)_{v,c}(\hat{A}Z)_{u,c}\right] = \mathbb{E}\left[(\sum_{k \in \mathcal{V}} \hat{A}_{v,k}\mathbf{1}_{\{Z_{k,:}=e_c^T\}})(\sum_{k' \in \mathcal{V}} \hat{A}_{u,k'}\mathbf{1}_{\{Z_{k',:}=e_c^T\}})\right]$$

$$= \mathbb{E}\left[(\sum_{k \in \mathcal{V}} \hat{A}_{v,k}\mathbf{1}_{\{Z_{k,:}=e_c^T\}})\right]\mathbb{E}\left[(\sum_{k' \in \mathcal{V}} \hat{A}_{u,k'}\mathbf{1}_{\{Z_{k',:}=e_c^T\}})\right]$$

Thus,

$$\mathbb{E}\left[S(\hat{A}, Z)_{v,u}\right] = \mathbb{E}\left[< (\hat{A}Z)_{v,:}, (\hat{A}Z)_{u,:} >\right] = \sum_c \mathbb{E}\left[(\sum_{k \in \mathcal{V}} \hat{A}_{v,k}\mathbf{1}_{\{Z_{k,:}=e_c^T\}})\right]\mathbb{E}\left[(\sum_{k' \in \mathcal{V}} \hat{A}_{u,k'}\mathbf{1}_{\{Z_{k',:}=e_c^T\}})\right]$$

$$= \begin{cases} \left(\frac{hd+1}{d+1}\right)^2 + \frac{((1-h)d)^2}{(C-1)(d+1)^2}, & u, v \text{ are in the same class} \\ \frac{2(hd+1)(1-h)d}{(C-1)(d+1)^2} + \frac{(C-2)(1-h)^2 d^2}{(C-1)^2(d+1)^2}, & u, v \text{ are in different classes} \end{cases}$$

For nodes $u_1, u_2$, and $v$, where $Z_{u_1,:} = Z_{v,:}$ and $Z_{u_2,:} \neq Z_{v,:}$,

$$g(h) \equiv \mathbb{E}\left[S(\hat{A}, Z)_{v,u_1}\right] - \mathbb{E}\left[S(\hat{A}, Z)_{v,u_2}\right] \tag{15}$$

$$= \frac{(C-1)^2(hd+1)^2 + (C-1)\left[(1-h)d\right]^2 - (C-1)\left(2(hd+1)(1-h)d\right) - (C-2)\left[(1-h)d\right]^2}{(C-1)^2(d+1)^2}$$

$$= \left(\frac{(C-1)(hd+1) - (1-h)d}{(C-1)(d+1)}\right)^2$$

Setting $g(h) = 0$, we obtain the optimal $h$:

$$h = \frac{d+1-C}{Cd} \tag{16}$$

For the data generation process in the synthetic experiments, we fix $d_{\text{intra}}$, then $d = d_{\text{intra}}/h$, which is a function of $h$. We change $d$ in equation 16 to $d_{\text{intra}}/h$, leading to

$$h = \frac{d_{\text{intra}}/h + 1 - C}{Cd_{\text{intra}}/h} \tag{17}$$

It is easy to observe that $h$ satisfying equation 17 still makes $g(h) = 0$, when $d$ in $g(h)$ is replaced by $d_{\text{intra}}/h$. From equation 17 we obtain the optimal $h$ in terms of $d_{\text{intra}}$:

$$h = \frac{d_{\text{intra}}}{Cd_{\text{intra}} + C - 1}$$

$\square$

## D.1 An Extension of Proposition 1

Base on the definition of aggregation similarity, we have

$$S_{\text{agg}}\left(S(\hat{A}, Z)\right) = \frac{\left|\left\{v \mid \text{Mean}_u\left(\{S(\hat{A}, Z)_{v,u}|Z_{u,:} = Z_{v,:}\}\right) \geq \text{Mean}_u\left(\{S(\hat{A}, Z)_{v,u}|Z_{u,:} \neq Z_{v,:}\}\right)\right\}\right|}{|\mathcal{V}|}$$

$$= \frac{\sum\limits_{v \in \mathcal{V}} \mathbf{1}_{\left\{ \mathrm{Mean}_u \left( \{S(\hat{A},Z)_{v,u} | Z_{u,:} = Z_{v,:}\} \right) \geq \mathrm{Mean}_u \left( \{S(\hat{A},Z)_{v,u} | Z_{u,:} \neq Z_{v,:}\} \right) \right\}}}{|\mathcal{V}|}$$

Then,

$$\mathbb{E}\left( S_{\mathrm{agg}}\left( S(\hat{A},Z) \right) \right) = \mathbb{E}\left( \frac{\sum\limits_{v \in \mathcal{V}} \mathbf{1}_{\left\{ \mathrm{Mean}_u \left( \{S(\hat{A},Z)_{v,u} | Z_{u,:} = Z_{v,:}\} \right) \geq \mathrm{Mean}_u \left( \{S(\hat{A},Z)_{v,u} | Z_{u,:} \neq Z_{v,:}\} \right) \right\}}}{|\mathcal{V}|} \right)$$

$$= \frac{\sum\limits_{v \in \mathcal{V}} \mathbb{P}\left( \mathrm{Mean}_u \left( \{S(\hat{A},Z)_{v,u} | Z_{u,:} = Z_{v,:}\} \right) \geq \mathrm{Mean}_u \left( \{S(\hat{A},Z)_{v,u} | Z_{u,:} \neq Z_{v,:}\} \right) \right)}{|\mathcal{V}|}$$

$$= \mathbb{P}\left( \mathrm{Mean}_u \left( \{S(\hat{A},Z)_{v,u} | Z_{u,:} = Z_{v,:}\} \right) - \mathrm{Mean}_u \left( \{S(\hat{A},Z)_{v,u} | Z_{u,:} \neq Z_{v,:}\} \right) \geq 0 \right)$$

Consider the random variable

$$RV = \mathrm{Mean}_u \left( \{S(\hat{A},Z)_{v,u} | Z_{u,:} = Z_{v,:}\} \right) - \mathrm{Mean}_u \left( \{S(\hat{A},Z)_{v,u} | Z_{u,:} \neq Z_{v,:}\} \right)$$

Since $RV$ is symmetrically distributed and under the conditions in proposition 1, its expectation is $\mathbb{E}[RV] = g(h)$ as showed in equation 15. Since the minimum of $g(h)$ is 0 and $RV$ is symmetrically distributed, we have $\mathbb{P}(RV \geq 0) \geq 0.5$ and this can explain why $H_{\mathrm{agg}}(\mathcal{G})$ is always greater than 0.5 in many real-world tasks.

## E   PROOF OF THEOREM 1

*Proof.* Define $W_v^c = (\hat{A}Z)_{v,c}$. Then,

$$W_v^c = \sum_{k \in \mathcal{V}} \hat{A}_{v,k} \mathbf{1}_{\{Z_{k,:} = e_c^T\}} \in [0,1], \quad \sum_{c=1}^{C} W_v^c = 1$$

Note that

$$S(I - \hat{A}, Z) = (I - \hat{A})ZZ^T(I - \hat{A})^T = ZZ^T + \hat{A}ZZ^T\hat{A}^T - \hat{A}ZZ^T - ZZ^T\hat{A}^T \qquad (18)$$

For any node $v$, let the class $v$ belongs to be denoted by $c_v$. For two nodes $v, u$, if $Z_{v,:} \neq Z_{u,:}$, we have

$$(ZZ^T)_{v,u} = 0$$

$$(\hat{A}ZZ^T\hat{A}^T)_{v,u} = \sum_{c=1}^{C} W_v^c W_u^c$$

$$(\hat{A}ZZ^T)_{v,u} = W_v^{c_u}$$

$$(ZZ^T\hat{A}^T)_{v,u} = (\hat{A}ZZ^T)_{u,v} = W_u^{c_v}$$

Then, from equation 18 it follows that

$$(S(I - \hat{A}, Z))_{v,u} = \sum_{c=1}^{C} W_v^c W_u^c - W_v^{c_u} - W_u^{c_v}$$

When $C = 2$,

$$S(I - \hat{A}, Z)_{v,u} = W_v^{c_u}(W_u^{c_u} - 1) + W_u^{c_v}(W_v^{c_v} - 1) \leq 0$$

If $Z_{v,:} = Z_{u,:}$, *i.e.*, $c_v = c_u$, we have

$$(ZZ^T)_{v,u} = 1$$

$$(\hat{A}ZZ^T\hat{A}^T)_{v,u} = \sum_{c=1}^{C} W_v^c W_u^c$$

$$(\hat{A}ZZ^T)_{v,u} = W_v^{c_v}$$
$$(ZZ^T\hat{A}^T)_{v,u} = (\hat{A}ZZ^T)_{u,v} = W_u^{c_u} = W_u^{c_v}$$

Then, from equation 18 it follows that

$$S(I - \hat{A}, Z)_{v,u} = 1 + \sum_{c=1}^{C} W_v^c W_u^c - W_v^{c_v} - W_u^{c_v}$$
$$= \sum_{c=1, c \neq c_v}^{C} W_v^c W_u^c + 1 + W_v^{c_v} W_u^{c_v} - W_v^{c_v} - W_u^{c_v}$$
$$= \sum_{c=1, c \neq c_v}^{C} W_v^c W_u^c + (1 - W_v^{c_v})(1 - W_u^{c_v}) \geq 0$$

Thus, if $C = 2$, for any $v \in \mathcal{V}$, if $Z_{u,:} \neq Z_{v,:}$, we have $S(I - \hat{A}, Z)_{v,u} \leq 0$; if $Z_{u,:} = Z_{v,:}$, we have $S(I - \hat{A}, Z)_{v,u} \geq 0$. Apparently, the two conditions in equation 9 are satisfied. Thus $v$ is diversification distinguishable and $\mathrm{DD}_{\hat{A},X}(\mathcal{G}) = 1$. The theorem is proved. □

# F    MODEL COMPARISON ON SYNTHETIC GRAPHS

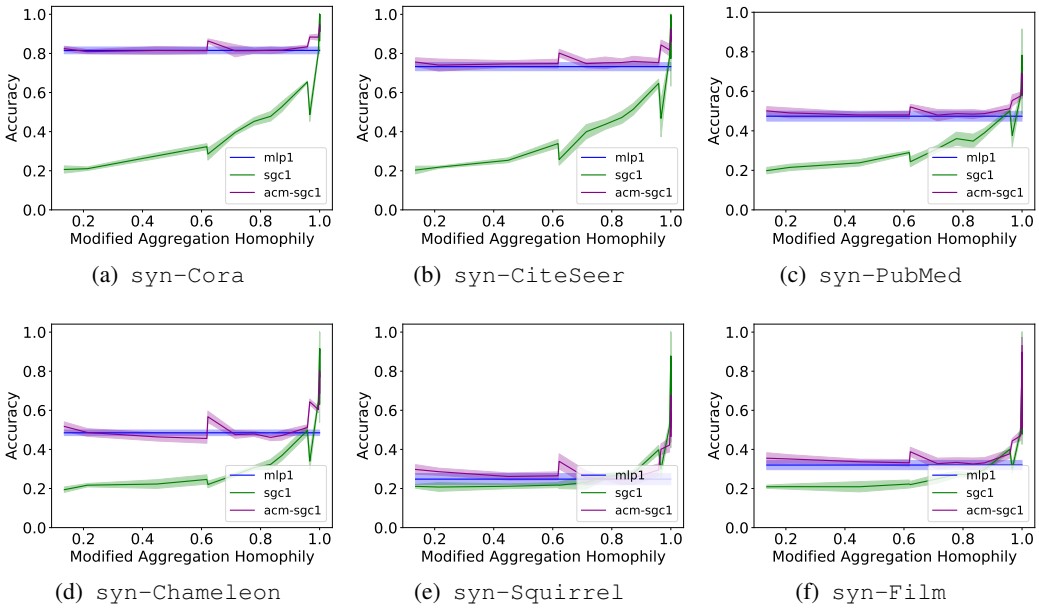

Figure 6: Comparison of test accuracy (mean ± std) of MLP-1, SGC-1 and ACM-SGC-1 on synthetic datasets

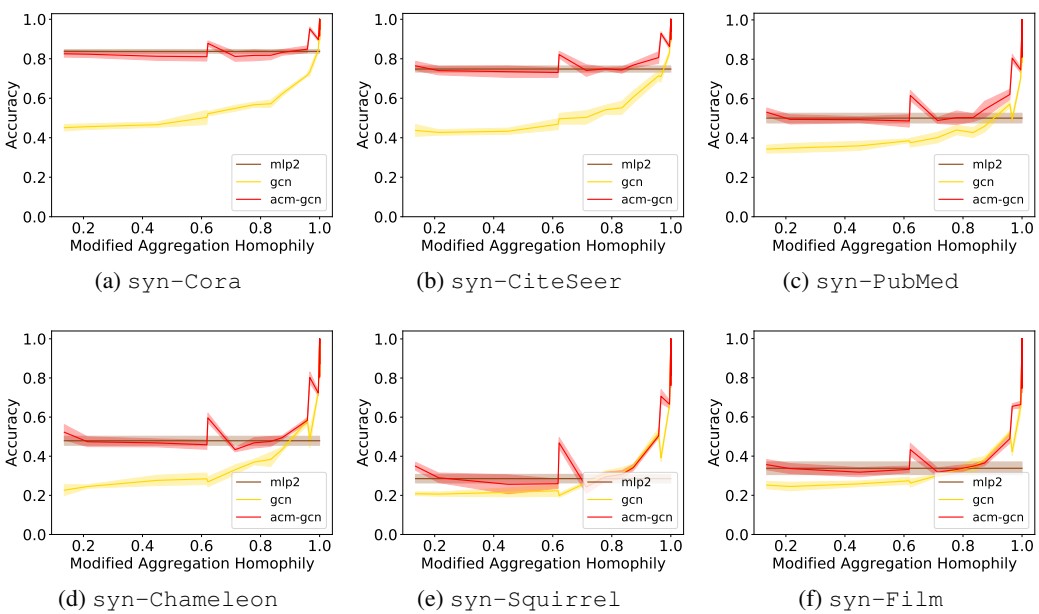

Figure 7: Comparison of test accuracy (mean $\pm$ std) of MLP-2, GCN and ACM-GCN on synthetic datasets

In order to separate the effects of nonlinearity and graph structure, we compare sgc with 1 hop (sgc-1) with MLP-1 (linear model). For GCN which includes nonlinearity, we use MLP-2 as its corresponding graph-agnostic baseline model. We train the above GNN models, graph-agnostic baseline models and ACM-GNN models on all synthetic datasets and plot the mean test accuracy with standard deviation on each dataset. From Figure 6 and Figure 7, we can see that on each $H_{\text{agg}}^M(\mathcal{G})$ level, ACM-GNNs will not underperform baseline GNNs and the graph-agnostic models. But when $H_{\text{agg}}^M(\mathcal{G})$ is small, baseline GNNs will be outperformed by graph-agnostic models by a large margin. This demonstrate the advantage of the ACM framework. The curves will almost be the same for ACMII framework.

## G  DISCUSSION OF THE LIMITATIONS OF DIVERSIFICATION OPERATION

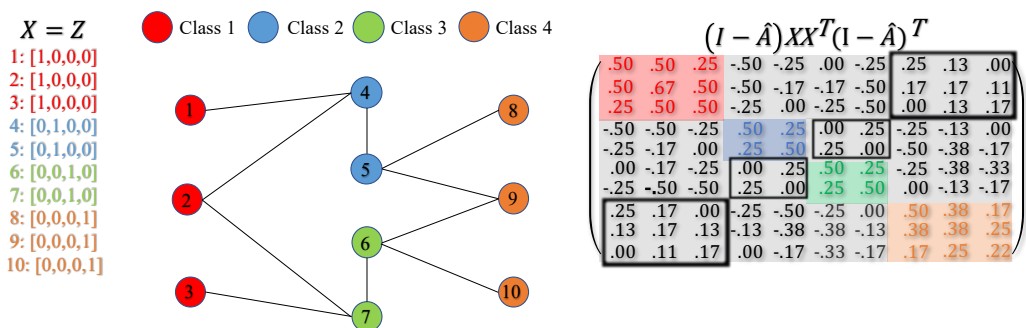

Figure 8: Example of the case (the area in black box) that HP filter does not work well for harmful heterophily

From the black box area of $S(I - \hat{A}, X)$ in the example in Figure 8 we can see that nodes in class 1 and 4 assign non-negative weights to each other although there is no edge between them; nodes in class 2 and 3 assign non-negative weights to each other as well. This is because the surrounding differences of class 1 are similar as class 4, so are class 2 and 3. In real-world applications, when

nodes in several small clusters connect to a large cluster, the surrounding differences of the nodes in the small clusters will become similar. In such case, HP filter are not able to distinguish the nodes from different small clusters.

## H  THE SIMILARITY, HOMOPHILY AND $\mathrm{DD}_{\hat{A},X}(\mathcal{G})$ METRICS AND THEIR ESTIMATIONS

**Additional Metrics**   There are three key factors that influence the performance of GNNs in real-world tasks: labels, features and graph structure. The (modified) aggregation homophily tries to investigate how the consistency of graph structure and labels will influence the performance of GNNs with features being fixed. And their correlation is verified through the synthetic experiments.

Besides graph-label consistency, we need to consider feature-label consistency and aggregated-feature-label consistency as well to fully investigate the performance of NNs and GNNs. With aggregation similarity score of the features $S_{\mathrm{agg}}\left(S(I, X)\right)$ and aggregated features $S_{\mathrm{agg}}\left(S(\hat{A}, X)\right)$ listed in table 8, our methods open up a new perspective on analyzing and comparing the performance of graph-agnostic models and graph-aware models in real-world tasks. Here are 2 examples.

Example 1: People observe that GCN (graph-aware model) underperforms MLP-2 (graph-agnostic model) on *Cornell, Wisconsin, Texas, Film*. Based on the aggregation homophily, the graph structure is not the main cause of the performance degradation. And from Table 6 we can see that the $S_{\mathrm{agg}}\left(S(\hat{A}, X)\right)$ for the above 4 datasets are lower than their corresponding $S_{\mathrm{agg}}\left(S(I, X)\right)$, which implies that it is the aggregated-feature-label inconsistency that causes the performance degradation.

Example 2: According to the widely used analysis based on node or edge homophily, the graph structure of *Chameleon*, and *Squirrel* are heterophilous and bad for GNNs. But in practice, GCN outperforms MLP-2 on those 2 datasets which means the additional graph information is helpful for node classification instead of being harmful. Traditional homophily metrics fail to explain such phenomenon but our method can give an explanation from different angles: For Chameleon, its modified aggregation homophily is not low and its $S_{\mathrm{agg}}\left(S(\hat{A}, X)\right)$ is higher than its $S_{\mathrm{agg}}\left(S(I, X)\right)$ which means its graph-label consistency and aggregated-feature-label consistency help the graph-aware model obtain the performance gain; for Squirrel, its modified aggregation homophily is low but its $S_{\mathrm{agg}}\left(S(\hat{A}, X)\right)$ is higher than its $S_{\mathrm{agg}}\left(S(I, X)\right)$ which means although its graph-label consistency is bad but the aggregated-feature-label consistency is the key factor to help the graph-aware model perform better.

We also need to point out that (modified) aggregation similarity score, $S_{\mathrm{agg}}\left(S(\hat{A}, X)\right)$ and $S_{\mathrm{agg}}\left(S(I, X)\right)$ are not deciding or threshold values because they do not consider the nonlinear structure in the features. In practice, a low score does not tell us the GNN models will definitely perform bad.

| | Cornell | Wisconsin | Texas | Film | Chameleon | Squirrel | Cora | CiteSeer | PubMed |
|---|---|---|---|---|---|---|---|---|---|
| $H_{\mathrm{agg}}(\mathcal{G})$ | 0.9016 | 0.8884 | 0.847 | 0.8411 | 0.805 | 0.6783 | 0.9952 | 0.9913 | 0.9716 |
| $S_{\mathrm{agg}}\left(S(\hat{A}, X)\right)$ | 0.8251 | 0.7769 | 0.6557 | 0.5118 | 0.8292 | 0.7216 | 0.9439 | 0.9393 | 0.8623 |
| $S_{\mathrm{agg}}\left(S(I, X)\right)$ | 0.9672 | 0.8287 | 0.9672 | 0.5405 | 0.7931 | 0.701 | 0.9103 | 0.9315 | 0.8823 |
| $DD_{\hat{A},X}(\mathcal{G})$ | 0.3497 | 0.6096 | 0.459 | 0.3279 | 0.3109 | 0.2711 | 0.2681 | 0.4124 | 0.1889 |
| $\hat{H}_{\mathrm{agg}}(\mathcal{G})$ | $0.9046 \pm 0.0282$ | $0.9147 \pm 0.0260$ | $0.8596 \pm 0.0299$ | $0.8451 \pm 0.0041$ | $0.8041 \pm 0.0078$ | $0.6788 \pm 0.0077$ | $0.9959 \pm 0.0011$ | $0.9907 \pm 0.0015$ | $0.9724 \pm 0.0015$ |
| $\hat{S}_{\mathrm{agg}}\left(S(\hat{A}, X)\right)$ | $0.8266 \pm 0.0526$ | $0.8280 \pm 0.0351$ | $0.6835 \pm 0.0498$ | $0.5345 \pm 0.0421$ | $0.8433 \pm 0.0070$ | $0.7352 \pm 0.0132$ | $0.9487 \pm 0.0023$ | $0.9451 \pm 0.0038$ | $0.8626 \pm 0.0021$ |
| $\hat{S}_{\mathrm{agg}}\left(S(I, X)\right)$ | $0.9752 \pm 0.0174$ | $0.8680 \pm 0.0270$ | $0.9661 \pm 0.0336$ | $0.5438 \pm 0.0184$ | $0.8257 \pm 0.0050$ | $0.7472 \pm 0.0089$ | $0.9204 \pm 0.0044$ | $0.9441 \pm 0.0036$ | $0.8835 \pm 0.0019$ |
| $DD_{\hat{A},X}(\mathcal{G})$ | $0.3936 \pm 0.0663$ | $0.6073 \pm 0.0436$ | $0.4817 \pm 0.0762$ | $0.3300 \pm 0.0136$ | $0.3329 \pm 0.0151$ | $0.3021 \pm 0.0101$ | $0.3198 \pm 0.0225$ | $0.4424 \pm 0.0136$ | $0.1919 \pm 0.0046$ |

Table 8: Additional metrics and their estimations with only training labels (mean $\pm$ std)

Furthermore, in most real-world applications, not all labels are available to calculate the dataset statistics. In this section, We randomly split the data into 60%/20%/20% for training/validation/test, and only use the training labels for the estimation of the statistics. We repeat each estimation for 10 times and report the mean with standard deviation. The results are shown in table 8.

**Analysis**   From the reported results we can see that the estimations are accurate and the errors are within the acceptable range, which means the proposed metrics and similarity scores can be accurately estimated with a subset of labels and this is important for real-world applications.

## I   EXPERIMENTS ON FIXED SPLITS PROVIDED BY (PEI ET AL., 2020)

See table 9 for the results and table 10 the optimal searched hyperparameters.

| | Cornell | Wisconsin | Texas | Film | Chameleon | Squirrel | Cora | CiteSeer | PubMed | Rank |
|---|---|---|---|---|---|---|---|---|---|---|
| GPRGNN | $78.11 \pm 6.55$ | $82.55 \pm 6.23$ | $81.35 \pm 5.32$ | $35.16 \pm 0.9$ | $62.59 \pm 2.04$ | $46.31 \pm 2.46$ | $87.95 \pm 1.18$ | $77.13 \pm 1.67$ | $87.54 \pm 0.38$ | 8.22 |
| H2GCN | $82.70 \pm 5.28$ | $87.65 \pm 4.98$ | $84.86 \pm 7.23$ | $35.70 \pm 1.00$ | $60.11 \pm 2.15$ | $36.48 \pm 1.86$ | $87.87 \pm 1.20$ | $77.11 \pm 1.57$ | $89.49 \pm 0.38$ | 6.78 |
| FAGCN | $76.76 \pm 5.87$ | $79.61 \pm 1.58$ | $76.49 \pm 2.87$ | $34.82 \pm 1.35$ | $46.07 \pm 2.11$ | $30.83 \pm 0.69$ | $\mathbf{88.05 \pm 1.57}$ | $77.07 \pm 2.05$ | $88.09 \pm 1.38$ | 9.56 |
| Geom-GCN* | $60.54 \pm 3.67$ | $64.51 \pm 3.66$ | $66.76 \pm 2.72$ | $31.59 \pm 1.15$ | $60.00 \pm 2.81$ | $38.15 \pm 0.92$ | $85.35 \pm 1.57$ | $\mathbf{78.02 \pm 1.15}$ | $89.95 \pm 0.47$ | 9.22 |
| ACM-SGC-1 | $82.43 \pm 5.44$ | $86.47 \pm 3.77$ | $81.89 \pm 4.53$ | $35.49 \pm 1.06$ | $63.99 \pm 1.66$ | $45.00 \pm 1.4$ | $86.9 \pm 1.38$ | $76.73 \pm 1.59$ | $88.49 \pm 0.51$ | 8.44 |
| ACM-SGC-2 | $82.43 \pm 5.44$ | $86.47 \pm 3.77$ | $81.89 \pm 4.53$ | $36.04 \pm 0.83$ | $59.21 \pm 2.22$ | $40.02 \pm 0.96$ | $87.69 \pm 1.07$ | $76.59 \pm 1.69$ | $89.01 \pm 0.6$ | 8.22 |
| ACM-GCN | $85.14 \pm 6.07$ | $\mathbf{88.43 \pm 3.22}$ | $\mathbf{87.84 \pm 4.4}$ | $36.28 \pm 1.09$ | $66.93 \pm 1.85$ | $\mathbf{54.4 \pm 1.88}$ | $87.91 \pm 0.95$ | $77.32 \pm 1.7$ | $\mathbf{90.00 \pm 0.52}$ | **2.33** |
| ACM-Snowball-2 | $85.41 \pm 5.43$ | $87.06 \pm 2$ | $87.57 \pm 4.86$ | $\mathbf{36.89 \pm 1.18}$ | $\mathbf{67.08 \pm 2.04}$ | $52.5 \pm 1.49$ | $87.42 \pm 1.09$ | $76.41 \pm 1.38$ | $89.89 \pm 0.57$ | 4.11 |
| ACM-Snowball-3 | $83.24 \pm 5.38$ | $86.67 \pm 4.37$ | $\mathbf{87.84 \pm 3.87}$ | $36.82 \pm 0.94$ | $66.91 \pm 1.73$ | $53.31 \pm 1.88$ | $87.1 \pm 0.93$ | $75.91 \pm 1.57$ | $89.81 \pm 0.43$ | 5.22 |
| ACMII-GCN | $\mathbf{85.95 \pm 5.64}$ | $87.45 \pm 3.74$ | $86.76 \pm 4.75$ | $36.16 \pm 1.11$ | $66.91 \pm 2.55$ | $51.85 \pm 1.38$ | $88.01 \pm 1.08$ | $77.15 \pm 1.45$ | $89.89 \pm 0.43$ | 3.22 |
| ACMII-Snowball-2 | $85.68 \pm 5.93$ | $87.45 \pm 2.8$ | $86.76 \pm 4.43$ | $36.55 \pm 1.24$ | $66.49 \pm 1.75$ | $50.15 \pm 1.4$ | $87.57 \pm 0.86$ | $76.92 \pm 1.45$ | $89.84 \pm 0.48$ | 4.67 |
| ACMII-Snowball-3 | $82.7 \pm 4.86$ | $85.29 \pm 4.23$ | $85.41 \pm 6.42$ | $36.49 \pm 1.41$ | $66.86 \pm 1.74$ | $48.87 \pm 1.23$ | $87.16 \pm 1.01$ | $76.18 \pm 1.55$ | $89.73 \pm 0.52$ | 7.00 |

Table 9: Experimental results on fixed splits provided by Pei et al. (2020): average test accuracy $\pm$ standard deviation on 9 real-world benchmark datasets. The best results are highlighted. Results of Geom-GCN, H$_2$GCN and GPRGNN are from Pei et al. (2020); Zhu et al. (2020b); Lingam et al. (2021); results on the rest models are run by ourselves and the hyperparameter searching range is the same as table 5.

## J   A DETAILED EXPLANATION OF THE DIFFERENCES BETWEEN ACM-GNNS, ACM(II)-GNNS AND GPRGNN, FAGCN

- Difference with GPRGNN (Chien et al., 2021): To explain how the channel mixing mechanism makes ACM-GNNs and ACMII-GNNs different from the learning mechanism in GPRGNN, we first rewrite GPRGNN as $\mathbf{Z} = \sum_{k=0}^{K} \gamma_k \mathbf{H}^{(k)} = \sum_{k=0}^{K} \gamma_k I \mathbf{H}^{(k)} = \sum_{k=0}^{K} diag(\gamma_k, \gamma_k, \ldots, \gamma_k) \mathbf{H}^{(k)}$. The node-wise channel mixing mechanism in GPRGNN form is $\mathbf{Z} = \sum_{k=0}^{K} diag(\gamma_k^1, \gamma_k^2, \ldots, \gamma_k^N) \mathbf{H}^{(k)}$, where $N$ is the number of nodes and $\gamma_k^i, i = 1, \ldots, N$ are learnable parametric mixing weights for different channels. ACM and ACMII allow GNNs to learn more diverse parameters in diagonal than GPRGNN and thus, have stronger expressive power than GPRGNN.

- Difference with FAGCN (Bo et al., 2021): instead of using a fixed $\hat{A}$, FAGCN learns a new filter $\hat{A}'$ based on $\hat{A}$ in a similar way as GAT (Velickovic et al., 2017). Some people may take $\hat{A}'$ as a mixing matrix and think it is similar to our channel mixing mechanism, but $\hat{A}'$ is essentially a learnable aggregator or LP filter for LP channel, which is far different from channel mixing. And $\hat{A}'$ can be decomposed into $\hat{A}' = \hat{A}'_1 - \hat{A}'_2$, where $\hat{A}'_1$ and $-\hat{A}'_2$ represents positive and negative edge (propagation) information respectively. In our paper, we are not discussing the advantages of using the learned filter $\hat{A}'$ over the fixed filter $\hat{A}$, we are comparing the models with and without channel mixing mechanism. We believe the empirical results on real-world tasks in table 4 and table 9 is the best way to compare the models with fixed filter and node-wise channel mixing and the models with learned filter but without channel mixing

| Datasets | Models\Hyperparameters | lr | weight_decay | dropout | hidden | results | std | average epoch time/average total time |
|---|---|---|---|---|---|---|---|---|
| Cornell | ACM-SGC-1 | 0.01 | 5.00E-06 | 0 | 64 | 82.43 | 5.44 | 5.37ms/23.05s |
|  | ACM-SGC-2 | 0.01 | 5.00E-06 | 0 | 64 | 82.43 | 5.44 | 5.93ms/25.66s |
|  | ACM-GCN | 0.05 | 5.00E-04 | 0.5 | 64 | 85.14 | 6.07 | 8.04ms/1.67s |
|  | ACMII-GCN | 0.1 | 1.00E-04 | 0 | 64 | 85.95 | 5.64 | 7.83ms/2.66s |
|  | FAGCN | 0.01 | 1.00E-04 | 0.6 | 64 | 76.76 | 5.87 | 8.80ms/7.67s |
|  | ACM-Snowball-2 | 0.05 | 5.00E-03 | 0.3 | 64 | 85.41 | 5.43 | 11.50ms/2.35s |
|  | ACM-Snowball-3 | 0.05 | 5.00E-03 | 0.2 | 64 | 83.24 | 5.38 | 15.06ms/3.12s |
|  | ACMII-Snowball-2 | 0.1 | 5.00E-03 | 0.2 | 64 | 85.68 | 5.93 | 12.63ms/2.58s |
|  | ACMII-Snowball-3 | 0.05 | 5.00E-03 | 0.2 | 64 | 82.7 | 4.86 | 14.59ms/3.06s |
| Wisconsin | ACM-SGC-1 | 0.1 | 5.00E-06 | 0 | 64 | 86.47 | 3.77 | 5.07ms/14.07s |
|  | ACM-SGC-2 | 0.1 | 5.00E-06 | 0 | 64 | 86.47 | 3.77 | 5.30ms/16.05s |
|  | ACM-GCN | 0.05 | 1.00E-05 | 0.4 | 64 | 88.43 | 3.22 | 8.04ms/1.66s |
|  | ACMII-GCN | 0.01 | 5.00E-05 | 0.1 | 64 | 87.45 | 3.74 | 8.40ms/2.19s |
|  | FAGCN | 0.01 | 5.00E-05 | 0.5 | 64 | 79.61 | 1.59 | 8.61ms/5.84s |
|  | ACM-Snowball-2 | 0.01 | 1.00E-03 | 0.4 | 64 | 87.06 | 2 | 12.51ms/2.60s |
|  | ACM-Snowball-3 | 0.01 | 1.00E-02 | 0.1 | 64 | 86.67 | 4.37 | 14.92ms/3.15s |
|  | ACMII-Snowball-2 | 0.01 | 5.00E-04 | 0.1 | 64 | 87.45 | 2.8 | 11.96ms/2.63s |
|  | ACMII-Snowball-3 | 0.01 | 5.00E-03 | 0.5 | 64 | 85.29 | 4.23 | 14.87ms/3.10s |
| Texas | ACM-SGC-1 | 0.01 | 1.00E-05 | 0 | 64 | 81.89 | 4.53 | 5.34ms/19.00s |
|  | ACM-SGC-2 | 0.05 | 5.00E-04 | 0 | 64 | 81.89 | 4.53 | 5.50ms/9.26s |
|  | ACM-GCN | 0.05 | 5.00E-04 | 0.5 | 64 | 87.84 | 4.4 | 9.62ms/1.99s |
|  | ACMII-GCN | 0.01 | 1.00E-03 | 0.1 | 64 | 86.76 | 4.75 | 9.98ms/2.22s |
|  | FAGCN | 0.01 | 1.00E-05 | 0 | 64 | 76.49 | 2.87 | 10.45ms/5.70s |
|  | ACM-Snowball-2 | 0.01 | 5.00E-03 | 0.2 | 64 | 87.57 | 4.86 | 11.56ms/2.45s |
|  | ACM-Snowball-3 | 0.01 | 5.00E-03 | 0.2 | 64 | 87.84 | 3.87 | 15.17ms/3.15s |
|  | ACMII-Snowball-2 | 0.01 | 1.00E-03 | 0.2 | 64 | 86.76 | 4.43 | 11.36ms/2.30 |
|  | ACMII-Snowball-3 | 0.01 | 5.00E-03 | 0.6 | 64 | 85.41 | 6.42 | 15.84ms/3.48s |
| Film | ACM-SGC-1 | 0.05 | 5.00E-04 | 0 | 64 | 35.49 | 1.06 | 5.39ms/1.17s |
|  | ACM-SGC-2 | 0.05 | 5.00E-04 | 0.1 | 64 | 36.04 | 0.83 | 13.22ms/3.31s |
|  | ACM-GCN | 0.01 | 5.00E-03 | 0 | 64 | 36.28 | 1.09 | 8.96ms/1.82s |
|  | ACMII-GCN | 0.01 | 5.00E-03 | 0 | 64 | 36.16 | 1.11 | 9.06ms/1.83s |
|  | FAGCN | 0.01 | 5.00E-05 | 0.4 | 64 | 34.82 | 1.35 | 15.60ms/2.51s |
|  | ACM-Snowball-2 | 0.01 | 1.00E-02 | 0 | 64 | 36.89 | 1.18 | 14.77ms/3.01s |
|  | ACM-Snowball-3 | 0.01 | 1.00E-02 | 0.2 | 64 | 36.82 | 0.94 | 16.57ms/3.36s |
|  | ACMII-Snowball-2 | 0.01 | 5.00E-03 | 0.1 | 64 | 36.55 | 1.24 | 12.76ms/2.57s |
|  | ACMII-Snowball-3 | 0.05 | 5.00E-03 | 0.3 | 64 | 36.49 | 1.41 | 16.51ms/3.49s |
| Chameleon | ACM-SGC-1 | 0.1 | 5.00E-06 | 0.9 | 64 | 63.99 | 1.66 | 5.92ms/1.74s |
|  | ACM-SGC-2 | 0.1 | 0.00E+00 | 0.9 | 64 | 59.21 | 2.22 | 8.84ms/1.78s |
|  | ACM-GCN | 0.05 | 5.00E-05 | 0.7 | 64 | 66.93 | 1.85 | 8.40ms/1.71s |
|  | ACMII-GCN | 0.05 | 5.00E-06 | 0.8 | 64 | 66.91 | 2.55 | 8.90ms/2.10s |
|  | FAGCN | 0.01 | 5.00E-05 | 0 | 64 | 46.07 | 2.11 | 16.90ms/7.94s |
|  | ACM-Snowball-2 | 0.01 | 1.00E-04 | 0.7 | 64 | 67.08 | 2.04 | 12.50ms/2.69s |
|  | ACM-Snowball-3 | 0.01 | 1.00E-05 | 0.8 | 64 | 66.91 | 1.73 | 16.12ms/4.91s |
|  | ACMII-Snowball-2 | 0.01 | 5.00E-05 | 0.8 | 64 | 66.49 | 1.75 | 12.65ms/3.42s |
|  | ACMII-Snowball-3 | 0.05 | 5.00E-05 | 0.7 | 64 | 66.86 | 1.74 | 17.60ms/4.06s |
| Squirrel | ACM-SGC-1 | 0.05 | 5.00E-06 | 0.9 | 64 | 45 | 1.4 | 6.10ms/2.18s |
|  | ACM-SGC-2 | 0.05 | 0.00E+00 | 0.9 | 64 | 40.02 | 0.96 | 35.75ms/9.62s |
|  | ACM-GCN | 0.05 | 5.00E-06 | 0.7 | 64 | 54.4 | 1.88 | 10.48ms/2.68s |
|  | ACMII-GCN | 0.05 | 5.00E-06 | 0.7 | 64 | 51.85 | 1.38 | 11.69ms/2.91s |
|  | FAGCN | 0 | 5.00E-03 | 0 | 64 | 30.86 | 0.69 | 10.90ms/13.91s |
|  | ACM-Snowball-2 | 0.01 | 1.00E-04 | 0.7 | 64 | 52.5 | 1.49 | 17.89ms/5.78s |
|  | ACM-Snowball-3 | 0.01 | 5.00E-05 | 0.7 | 64 | 53.31 | 1.88 | 22.60ms/7.53s |
|  | ACMII-Snowball-2 | 0.05 | 5.00E-05 | 0.6 | 64 | 50.15 | 1.4 | 16.95ms/3.45s |
|  | ACMII-Snowball-3 | 0.01 | 5.00E-04 | 0.6 | 64 | 48.87 | 1.23 | 23.52ms/4.94s |
| Cora | ACM-SGC-1 | 0.05 | 5.00E-05 | 0.7 | 64 | 86.9 | 1.38 | 4.99ms/2.40s |
|  | ACM-SGC-2 | 0.1 | 0 | 0.8 | 64 | 87.69 | 1.07 | 5.16ms/1.16s |
|  | ACM-GCN | 0.01 | 5.00E-05 | 0.6 | 64 | 87.91 | 0.95 | 8.41ms/1.84s |
|  | ACMII-GCN | 0.01 | 1.00E-04 | 0.6 | 64 | 88.01 | 1.08 | 8.59ms/1.96s |
|  | FAGCN | 0.02 | 1.00E-04 | 0.5 | 64 | 88.05 | 1.57 | 9.30ms/10.64s |
|  | ACM-Snowball-2 | 0.01 | 1.00E-03 | 0.5 | 64 | 87.42 | 1.09 | 12.54ms/2.72s |
|  | ACM-Snowball-3 | 0.01 | 5.00E-06 | 0.9 | 64 | 87.1 | 0.93 | 15.83ms/11.33s |
|  | ACMII-Snowball-2 | 0.01 | 1.00E-03 | 0.6 | 64 | 87.57 | 0.86 | 12.06ms/2.64s |
|  | ACMII-Snowball-3 | 0.01 | 5.00E-03 | 0.5 | 64 | 87.16 | 1.01 | 16.29ms/3.62s |
| CiteSeer | ACM-SGC-1 | 0.05 | 0.00E+00 | 0.7 | 64 | 76.73 | 1.59 | 5.24ms/1.14s |
|  | ACM-SGC-2 | 0.1 | 0.00E+00 | 0.8 | 64 | 76.59 | 1.69 | 5.14ms/1.03s |
|  | ACM-GCN | 0.01 | 5.00E-06 | 0.3 | 64 | 77.32 | 1.7 | 8.89ms/1.79s |
|  | ACMII-GCN | 0.01 | 5.00E-05 | 0.5 | 64 | 77.15 | 1.45 | 8.95ms/1.80s |
|  | FAGCN | 0.02 | 5.00E-05 | 0.4 | 64 | 77.07 | 2.05 | 10.05ms/5.69s |
|  | ACM-Snowball-2 | 0.01 | 5.00E-05 | 0 | 64 | 76.41 | 1.38 | 12.87ms/2.59s |
|  | ACM-Snowball-3 | 0.01 | 5.00E-06 | 0.9 | 64 | 75.91 | 1.57 | 17.40ms/11.92s |
|  | ACMII-Snowball-2 | 0.01 | 5.00E-03 | 0.5 | 64 | 76.92 | 1.45 | 13.10ms/2.94s |
|  | ACMII-Snowball-3 | 0.1 | 5.00E-05 | 0.9 | 64 | 76.18 | 1.55 | 17.47ms/5.88s |
| PubMed | ACM-SGC-1 | 0.05 | 5.00E-06 | 0.4 | 64 | 88.49 | 0.51 | 5.77ms/3.65s |
|  | ACM-SGC-2 | 0.05 | 5.00E-06 | 0.3 | 64 | 89.01 | 0.6 | 8.50ms/5.18s |
|  | ACM-GCN | 0.01 | 5.00E-05 | 0.4 | 64 | 90 | 0.52 | 8.99ms/2.51s |
|  | ACMII-GCN | 0.01 | 1.00E-04 | 0.3 | 64 | 89.89 | 0.43 | 9.70ms/2.57s |
|  | FAGCN | 0.01 | 1.00E-04 | 0 | 64 | 88.09 | 1.38 | 10.30ms/8.75s |
|  | ACM-Snowball-2 | 0.01 | 1.00E-03 | 0.3 | 64 | 89.89 | 0.57 | 15.05ms/3.11s |
|  | ACM-Snowball-3 | 0.01 | 5.00E-03 | 0.1 | 64 | 89.81 | 0.43 | 20.51ms/4.63s |
|  | ACMII-Snowball-2 | 0.01 | 5.00E-04 | 0.4 | 64 | 89.84 | 0.48 | 15.10ms/3.2s |
|  | ACMII-Snowball-3 | 0.01 | 1.00E-03 | 0.4 | 64 | 89.73 | 0.52 | 20.46ms/4.32s |

Table 10: Optimal Hyperparameters for FAGCN and ACM-GNNs on fixed splits

