# OpenReview forum: "Is Heterophily A Real Nightmare For Graph Neural Networks on Performing Node Classification?"
_ICLR.cc/2022/Conference — ICLR 2022 Submitted_

### Official Review · Reviewer_o6WB · 2021-11-01

**Correctness:** 4
**Technical Novelty And Significance:** 3
**Empirical Novelty And Significance:** 3
**Recommendation:** 8
**Confidence:** 3

**Main Review:**

The paper addresses a well known problem in the literature in a seemingly effective way. It is really well written, offering good background on the problem and going smoothly from the exposition of the problem and the limited view of existing approaches to the proposed ACM architecture to address it. Then, it performs very comprehensive experiments that demonstrate the effectiveness of the proposed scheme.

If I need to mention a weakness, this is the fact that the paper necessarily compresses a lot of discussion and description due to the conference constraints. For that reason, it misses a thorough analysis and comparison with the state of the art, as well as a more in-depth discussion of the experimental results and conclusions. Yet, the authors make the best of the available space to cover all necessary information.

**Summary Of The Paper:**

The paper starts from studying the question of how heterophily affects the learning effectiveness of Graph Neural Networks on node classification tasks and posits that heterophily may not be always detrimental to the task. Following this observation, it proposes a new architecture, called Adaptive Channel Mixing, which appropriately applies aggregation, diversification and identity channels in each GNN layer in order to tackle harmful heterophily. According to the experimental results, the proposed architecture is successful.

**Summary Of The Review:**

The paper proposes a new architecture, called Adaptive Channel Mixing, with the aim to address several cases of harmful heterophily in GNN node classification settings. The paper is well written and motivated and contains convincing experimentations. It could also have a big impact on GNN node classification practice and future research.

---

> ### Author Response · Authors · 2021-11-16
> **Response to Reviewer o6WB**
>
> Thanks so much for your positive comments. We will keep updating our paper.

---

### Official Review · Reviewer_vrWz · 2021-11-02

**Correctness:** 4
**Technical Novelty And Significance:** 3
**Empirical Novelty And Significance:** 4
**Recommendation:** 8
**Confidence:** 4

**Main Review:**

Pros:
(1) solid and deep analysis of the proposed metric and filterbank-based aggregation scheme.
(2) solid and comprehensive experiments to show the significance of the proposed ideas.

Cons:
(1) some necessary justifications are missing.
(2) presentation can be improved.

Detailed comments:
-In section 3.1, it is not reader friendly to mention "mean aggregation" before introducing Definition 1. The same comment can be applied to the beginning of Section 4: explicitly give the definition of diversification operation before discuss about it.
-Figure 2: how do you get the curves of (b) (c) (d) when you only manipulate H_edge?
-Some figures are not easily readable in b/w print.
-The three steps of ACM lacks explanations and justification. Specifically, in Steps 2 and 3, why using such a combination weight calculation scheme? The raw weights are estimated only based on the results from the corresponding filter. I feel it would be better to decide the weight for each channel based on all the filtered results. In step 3, why using W_{mix}? How do we interpret the temperature parameter here? Is the softmax in step 2 row-wise (need to explain it in order to improve readability)?
-Experiments: In Figure 4, how to interpret the relations between H_{agg}^M and the performance increase? It would be better to put important and necessary discussions in the main text.

-Minor typos: page 7, "called is", "a set of filterbank".

**Summary Of The Paper:**

This paper proposes a new aggregation-based homophily metric for assessing the homophily of a graph. This metric complements the current metrics by considering unharmful heterophily cases. The second contribution is a new filterbank framework which uses the diversification operation to fight harmful heterophily information.

**Summary Of The Review:**

This paper is well written and novel enough for this conference. The presentation can be improved and more justification can be added to further improve the paper.

---

> ### Author Response · Authors · 2021-11-16
> **Response to Reviewer vrWz**
>
>
> We appreciate your positive comments and valuable suggestions. Here are our response to your concerns and questions.
>
> **Q1**
>  "In Section 3.1, it is not reader friendly to mention "mean aggregation" before introducing Definition 1. The same comment can be applied to the beginning of Section 4: explicitly give the definition of diversification operation before discuss about it"
>
> **R1**
> Thanks for your suggestion. We will improve the presentation as you suggested.
>
> We mentioned mean aggregator in the last paragraph of Section 2.1 and assumed the readers can understand mean aggregation based on mean aggregator. But we will make it more clear and straight-forward in the revised version.
>
> We will add the definition of high-pass filter in Section 2.1 and give an explanation to the diversification operation in the revised version.
>
> **Q2**
>  "Figure 2: how do you get the curves of (b) (c) (d) when you only manipulate H_edge? -Some figures are not easily readable in b/w print"
>
> **R2**
>
> As mentioned in section 3.2, we generate graphs with different $H_\text{edge}$ and for each generated graph, we calculate their $H_\text{node},H_\text{class}, H_\text{agg}^M$.
>
> Unfortunately, we are unable to solve the b/w print problem. The only thing we can currently do to improve the readability is to enlarge the font size in the legend in the revised version.
>
> **Q3**
> -The three steps of ACM lacks explanations and justification. Specifically, (1) in Steps 2 and 3, why using such a combination weight calculation scheme? The raw weights are estimated only based on the results from the corresponding filter. I feel it would be better to decide the weight for each channel based on all the filtered results. In step 3, why using W_{mix}? (2) How do we interpret the temperature parameter here? (3) Is the softmax in step 2 row-wise (need to explain it in order to improve readability)?
>
>
> **R3**
> (1) The first line in step 2 is to extract nonlinear information from the filtered signal for each of the 3 channels.
> The second line in step 2 then uses $W_{mix}$ to learn which channel is
> important or not important for each node from the combined information, leading to the weight information for each channel and for each node, i.e., the three vectors $\alpha_L^l$, $\alpha_H^l$ and $\alpha_I^l$,
> whose $i$-th elements are the weights for the $i$-th node.
> These three vectors are then used as weights in defining the updated channel matrix $H^l$ in step 3.
>
>
> Actually, we have tried a lot of possible designs for the weight learning. The empirical evaluation on real-world tasks indicated this design performs the best.
>
> (2) The temperature $T$ is a hyperparameter and as mentioned in section 6.1, we set it to the number of channels, i.e., $T=3$. The goal is for normalization that makes $\tilde{\alpha}_L^l+\tilde{\alpha}_H^l+\tilde{\alpha}_I^l$ lie in [0,1].
> Mathematically this normalization is not needed.
> But due to some numerical issue with the optimization method,
> this normalization has an impact on the final result.
> We have tried $T=1,3,5$ and found $T=3$ works the best.
>
> (3) Yes, the softmax is row-wise. We mention it in the paragraph under equation (11).
> We will give explanations about the three steps in the revised version.
>
> **Q4**
> "Experiments: In Figure 4, how to interpret the relations between H_{agg}^M and the performance increase?"
>
> **R4**
> There is no direct and deterministic relation between $H_\text{agg}^M$ and the performance increase. The reason we put $H_{agg}^M$ in Figure 4 is just to show the calculated statistics for each dataset. Due to space limitation, the relation between $H_\text{agg}^M$ (together with other metrics) and GNN performance is discussed in Appendix H.

---

> > ### Comment · Reviewer_vrWz · 2021-11-19
> > **Not satisfied by some of the replies**
> >
> > Q2: I am wondering how you generate those curves (except for  H_edge) with smooth points along the x-axis.
> > Q3: I still think a better way is to decide the raw weight for each channel based on all the filtered results.  The authors' reply is not convincing by simply saying empirical results show the current design is the best choice.
> > Q4: It is not good to put the main result discussion in the appendix.

---

> > > ### Author Response · Authors · 2021-11-21
> > > **Response 2 to Reviewer vrWz**
> > >
> > >
> > >
> > >
> > > **Q1**: I am wondering how you generate those curves (except for H_edge) with smooth points along the x-axis.
> > >
> > > **R1**:
> > > We reorder the value of the metrics in ascend order for x-axis.
> > >
> > > Here is a simplified example. Suppose we generate graphs with $H_\text{edge}=0.1,0.5,0.9$, the test accuracy of GCN on the synthetic graphs are $0.8,0.5,0.9$.
> > > For the generated graphs, we calculate their $H_\text{agg}^M$, and suppose we get $H_\text{agg}^M=0.7,0.4,0.8$. Then we will draw the performance of GCN under $H_\text{agg}^M$ with x-axis $[0.4,0.7,0.8]$ and the corresponding reordered y-axis is $[0.5,0.8,0.9]$. Other metrics use the same process.
> > >
> > > **Q2**: I still think a better way is to decide the raw weight for each channel based on all the filtered results. The authors' reply is not convincing by simply saying empirical results show the current design is the best choice.
> > >
> > > **R2**:
> > > We thought that your "raw weight" comment in your last report was related to $W_{mix}$
> > > and perhaps it would not be an issue after we gave an explanation about the role of $W_{mix}$.
> > > We now realize we misunderstood it. Sorry about this.
> > > We have been trying to figure out what your suggestion meant exactly.
> > > Here is our guess:
> > > Replace the first line in Step 2 by the following lines:
> > >
> > > Construct the combined feature ${H}^{l}_\text{Comb} = [{H}^{l}_L,{H}^{l}_H,{H}^{l}_I]$ and use it in step 2  so that the raw weight for each channel is based on all the filtered results as follows,
> > >
> > > $\tilde{\alpha}_L^l = \sigma\left({H}^{l}_\text{Comb} \tilde{W}^{l}_L \right),\ \tilde{\alpha}_H^l = \sigma \left({H}^{l}_\text{Comb}\tilde{W}^{l}_H \right),\ \tilde{\alpha}_I^l = \sigma \left({H}^{l}_\text{Comb} \tilde{W}^{l}_I \right),\ \tilde{W}_L^{l-1},\ \tilde{W}_H^{l-1},\ \tilde{W}_I^{l-1} \in \mathbb{R}^{3F_l \times 1}$
> > > $ \left[{\alpha}_L^l, {\alpha}_H^l, {\alpha}_I^l \right] = \text{Softmax}\left((\left[\tilde{\alpha}_L^l,\tilde{\alpha}_H^l,\tilde{\alpha}_I^l\right]/T) W_\text{Mix}^l \right) \in \mathbb{R}^{N\times 3},\ T \in \mathbb{R} \text{ temperature},\ W_\text{Mix}^l \in \mathbb{R}^{3\times 3}; $
> > >
> > > Here are the performances of new methods (ACM-GCN (new), ACMII-GCN (new)) compared with the methods in our paper (ACM-GCN, ACMII-GCN):
> > >
> > > | Models\ Datasets  | Cornell | Wisconsin|Texas|Film|Squirrel|Cora|Citeseer|Pubmed|
> > > |:-:|:-:|:-:|:-:|:-:|:-:|:-:|:-:|:-:|
> > > |ACM-GCN | 94.75 $\pm$ 3.8 | 95.75 $\pm$ 2.03 | 94.92 $\pm$ 2.88 | 41.62 $\pm$ 1.15 | 69.04 $\pm$ 1.74 | 58.02 $\pm$ 1.86 | 88.62 $\pm$ 1.22 | 81.68 $\pm$ 0.97 | 90.66 $\pm$ 0.47 |
> > > |ACMII-GCN | 95.9 $\pm$ 1.83 | 96.62 $\pm$ 2.44 | 95.08 $\pm$ 2.07 | 41.84 $\pm$ 1.15 | 68.38 $\pm$ 1.36 | 54.53 $\pm$ 2.09 | 89.00 $\pm$ 0.72 | 81.79 $\pm$ 0.95 | 90.74 $\pm$ 0.5 |
> > > |ACM-GCN (new) | 95.08 $\pm$ 2.64 | 96.12 $\pm$ 1.31 | 94.92 $\pm$ 2.48 | 41.62 $\pm$ 1.34 | 68.82 $\pm$ 2.18 | 57.48 $\pm$ 1.68 | 88.59 $\pm$ 1.04 | 81.9 $\pm$ 1.27 | 90.75 $\pm$ 0.77  |
> > > |ACMII-GCN (new) | 93.93 $\pm$ 3.52 | 96 $\pm$ 2 | 94.59 $\pm$ 2.94 | 41.44 $\pm$ 1.18 | 68.53 $\pm$ 3.08 | 53.28 $\pm$ 1.08 | 88.75 $\pm$ 0.83 | 81.76 $\pm$ 1.05 | 90.58 $\pm$ 0.64 |
> > >
> > > From the results, we do not find significant differences between the frameworks with combined features and the ones without combined features in the feature extraction step. The reason we think is that the necessary nonlinear information from each channel is combined in $\left[\tilde{\alpha}_L^l,\tilde{\alpha}_H^l,\tilde{\alpha}_I^l\right]$ and $W_\text{Mix}^l$ is enough to learn to mix the combined weights from different channels. The learning of redundant information in the feature extraction step for each channel will not improve the performance.
> > > (A disadvantage of using the combined feature is that it increases the computational cost.)
> > >
> > > If our guess is not what you meant, could you please describe your suggestion mathematically?
> > >
> > > **Q3**: It is not good to put the main result discussion in the appendix.
> > >
> > > **R3**:
> > > Thanks for your advice. Due to the space limitation, we have no choice but to put some discussion to appendix and only leave the main results and conclusions in the main text. We will definitely move some of the discussion back to the main text if we are allowed to have one more page for the main paper.

---

### Official Review · Reviewer_ZQM7 · 2021-11-03

**Correctness:** 3
**Technical Novelty And Significance:** 2
**Empirical Novelty And Significance:** 3
**Recommendation:** 5
**Confidence:** 3

**Main Review:**

In this paper, the authors first analyze the potential drawbacks of existing heterophily metrics, then propose an aggregated heterophily metric aiming to better estimate the "harmful" heterophily and utilize diversification operation to address certain harmful heterophily cases. Based on the analysis, the authors propose an adaptive channel mixing (ACM) framework to improve GNN model performance. Several experiments have also been conducted to evaluate the proposed model. In general, the authors focus on an important problem and the paper is easy to follow. There are a few issues the authors need to address.

My first concern is that the heterophily analysis might be over-simplified. It is good to see that the authors introduce an example to show when traditional heterophily metrics may fail to provide a meaningful heterophily value. However, the analysis conducted has two major assumptions: binary classification and one-hop aggregation. Though I understand that certain assumption is required for most theoretical analysis, these two seem to be very strong given that multi-layer of GNNs are usually operated on graphs with multiple node labels. This reduces the significance of the analysis.

Some technical terms also need to be better explained or discussed. For example, the authors introduce high-pass (HP) and low-pass (LP) filters in Sec. 4 without introducing any motivation or rationales of applying them. I suggest the authors to better connect these filters with the analysis in Sec. 3 so that readers can better understand why in certain scenarios these two filters can help alleviate the heterophily problem. The authors briefly introduce some potential implementation of HP/LP filters in Sec. 4.2 while the exact implementation selected in the experiments is missing. I suggest the authors to explicitly indicate which filter implementations are selected and whether they're deterministic or learnable.

The ACM framework seems to be a weighted aggregation of the results after frequency filters. I suggested the authors to add an ablation study by comparing the ACM framework with some multi-head self-attention mechanisms to better understand where the gains are from.

**Summary Of The Paper:**

In this paper, the authors first analyze the potential drawbacks of existing heterophily metrics, then propose an aggregated heterophily metric aiming to better estimate the "harmful" heterophily and utilize diversification operation to address certain harmful heterophily cases. Based on the analysis, the authors propose an adaptive channel mixing (ACM) framework to improve GNN model performance. Several experiments have also been conducted to evaluate the proposed model.

**Summary Of The Review:**

Strength:
+ Focusing on an important problem
+ Paper is easy to follow

Weakness:
- Over-simplified theoretical analysis
- Some technical terms need to be explained / discussed
- Missing certain ablation studies

---

> ### Author Response · Authors · 2021-11-16
> **Response to Reviewer ZQM7**
>
> Thanks for you constructive suggestions and here are our response to your questions.
>
> **Q1**
> "My first concern is that the heterophily analysis might be over-simplified...the analysis conducted has two major assumptions: (1) binary classification and (2) one-hop aggregation..."
>
> **R1**
> (1) Our heterophily analyses are not all based on the binary classification. Specifically, the binary classification is used in the examples in Figure 1 and Figure 3 just to simplify the demonstration. The binary classification assumption is indeed used in Theorem 1 to illustrate the importance of diversification operation. But for multi-class problems, we have empirically and theoretically shown the advantages of the proposed metrics over the existing homophily metrics in  Section 3.2 and Appendix B. We also verify the effectiveness of the diversification operation on multi-class classification tasks in the ablation study .
>
> (2) Since the commonly used homophily metrics and heterophily analysis are all based on one-hop neighborhood as shown in Section 2.2, to follow the definition of homophily and make a fair comparison with the existing metrics, we define and analyze the new metric with one-hop aggregator. We will consider generalizing our analysis to multi-hop neighborhood in the future.
>
>
> **Q2**
> " ...the authors introduce high-pass (HP) and low-pass (LP) filters in Sec. 4 without introducing any motivation or rationales of applying them. I suggest the authors to better connect these filters with the analysis in Sec. 3..."
>
> **R2**
> Thanks for your suggestion. The motivation of analyzing diversification operation is based on the filterbank methods and experimental results. And in Section 4, we explain why diversification operation works for some harmful heterophily cases based on the method  developed in section 3. We will clarify the motivation and connection in the revised version.
>
> **Q3**
>
> "The authors briefly introduce some potential implementation of HP/LP filters in Sec. 4.2 while the exact implementation selected in the experiments is missing. I suggest the authors to explicitly indicate which filter implementations are selected and whether they're deterministic or learnable."
>
> **R3**
> Actually in Section 6.2 we use the random-walk renormalized affinity matrix $\hat{A}_{rw}$ as low-pass filter and $I - \hat{A}_\{rw\}$ as high-pass filter.
>
> The filters are deterministic and we will say it explicitly in the revised version.
>
> **Q4**
> "I suggested the authors to add an ablation study by comparing the ACM framework with some multi-head self-attention mechanisms to better understand where the gains are from."
>
> **R4**
>
> Thanks for this suggestion, but we cannot see the motivation and necessity to add multi-head self-attention in ablation study to understand the gains.
> If you meant
>
> (1) Multi-head self-attention should be used to replace the proposed node-wise channel mixing mechanism
>
> Our reply: The node-wise channel mixing mechanism is based on the argument that each node has different needs for the information from different channels as mentioned in Section 4.2. We do not find the motivation to do this ablation and how this replacement can help us understand the gains better. On the other hand, self-attention will hurt the efficiency of the algorithm and will not add any contribution to our paper.
>
> (2) Multi-head self-attention should be used to learn the edge weights in each channel
>
> Our reply: we do have the experimental results of GAT, 8-head GAT, ACM-GAT and 8-head ACM-GAT as follows, if these are what you are interested in.
>
> | Models\ Datasets  | Cornell | Wisconsin|Texas|Film|Squirrel|Cora|Citeseer|Pubmed|
> |:-:|:-:|:-:|:-:|:-:|:-:|:-:|:-:|:-:|
> |GAT (1-head)|58.92 $\pm$ 3.38|51.76 $\pm$ 6.80|60.81 $\pm$ 5.30|27.91 $\pm$ 0.93|28.42 $\pm$ 1.01|73.30 $\pm$ 1.17|70.15 $\pm$ 1.43|81.90 $\pm$ 0.59|
> |8-head GAT|58.92 $\pm$ 3.15|53.53 $\pm$ 4.89|60.81 $\pm$ 4.56|28.01 $\pm$ 1.19|28.41 $\pm$ 1.88|75.02 $\pm$ 1.00|71.33 $\pm$ 2.05|82.02 $\pm$ 0.70|
> |ACM-GAT (1-head)|71.35 $\pm$ 6.68|75.49 $\pm$ 5.06|72.97 $\pm$ 0.58|35.18 $\pm$ 1.13|30.44 $\pm$ 1.43|85.17 $\pm$1.46|74.79 $\pm$ 1.88|88.75 $\pm$ 0.62|
> |8-head ACM-GAT| **85.41 $\pm$ 3.06** |**83.53 $\pm$ 3.42**|**78.92 $\pm$ 7.91**|**35.55 $\pm$ 1.29**|**30.69 $\pm$ 1.23**|**86.58 $\pm$ 1.43**|**75.23 $\pm$ 1.63**|**89.49 $\pm$ 0.30**|
>
> In our opinion, the current ablation study in Section 6.1 has already clearly shown that the HP channel, identity channel and node-wise channel mixing mechanism are effective and helps us understand where the gains are from. We are not sure which part of our ablation study is unclear or hard to understand.

---

### Official Review · Reviewer_bk6o · 2021-11-04

**Correctness:** 2
**Technical Novelty And Significance:** 2
**Empirical Novelty And Significance:** 2
**Recommendation:** 3
**Confidence:** 3

**Main Review:**

Heterophily is an important factor for us to understand the performance of GNN on different graphs.  I think the paper found a good point that heterophily is not always harmful for all GNNs. This paper provides a new perspective to understand the problem, and designs a new metric based on the gradient analysis, which is interesting. The filterbank based method does solve the problem to some extent and gained very good performance.

However, 1. It is not clear why the gradient leads to the aggregation similarity score as the heterophily metric, and how does it impact the GNNs’ performance. There is also no direct connection with the later diversification analysis.
2. The diversification operation helps on heterophilous graphs also lacks of enough motivation and theoretic explanation. It seems there is only one example (Figure 3) to illustrate it. That is not enough. In addition, the diversification operation even cannot work well for all cases, especially for multi-class problems. This makes the value of this paper hurt a lot.
3. The filterbank based method is trivial and not new. Adding that the theoretic analysis of diversification operation or new metric is also not convincing, the novelty of this paper is incremental.


**Summary Of The Paper:**

The paper shows that not all cases of heterophily are harmful for GNNs with aggregation operations. Based on a backpropagation analysis on an SGC-style GNN, it provides a new metric based on similarity matrix which considers the influence of both graph structure and input features. Observing that the diversification operation is able to address some harmful heterophily cases, it proposes the Adaptive Channel Mixing GNN framework which combines a high-pass filter, a low-pass filter and an identity channel.

**Summary Of The Review:**

Good motivation and observations, but the solution is also not novel enough and lacks of theoretic justification.

---

> ### Author Response · Authors · 2021-11-16
> **Response to Reviewer bk6o**
>
> Thanks for you comments.
>
> **Q1**
> (1) It is not clear why the gradient leads to the aggregation similarity score as the heterophily metric, and how does it impact the GNNs’ performance. (2) There is also no direct connection with the later diversification analysis.
>
> **R1**
> (1) The performances of current NN and GNN models are almost all related to the gradient descent algorithm, and the core difference between graph-aware and graph-agnostic models is that the former leverage graph information in the backpropagation process, while the latter does not. Thus, analyzing how the graph information will influence the gradient can help us to intuitively understand how the updating directions of GNNs are different from NNs. Based on the gradient analysis, we derive the new homophily metrics.
>
> (2) We also study the effect of diversification operation from the gradient updating perspective with analysis of $S(I-\hat{A},X)$. From the observation in Figure 3, we find that the high-pass filter can help us to treat nodes from different classes as negative samples as we mentioned in section 4.1. This provides us a slightly different angle to analyze the high-pass filter and define diversification distinguishability.
>
> **Q2**
> The diversification operation helps on heterophilous graphs also lacks of enough motivation and theoretic explanation.
>
> **R2**
> The motivation of using diversification operation comes from the filterbank methods and our experimental experience. In Section 4, we use the $S(I-\hat{A},X)$ to illustrate how diversification operation works in some cases when aggregation operation does not work well. We also provide a theoretic explanation in Theorem 1 for the binary classification problem.
>
> **Q3**
> the diversification operation even cannot work well for all cases, especially for multi-class problems
>
> **R3**
> To our knowledge, no method can work well for all cases. We give an example of multi-class problem just to illustrate the limitation of diversification operation because we need to be honest for that. It does not mean our method cannot work well for all multi-class problems. In fact, the ablation study have shown the effectiveness of the diversification operation.
>
> Our main point is that the proposed method has advantages on dealing with heterophily problem over the uni-channel (low-pass channel) methods, which are used by almost all GNNs, although the additional diversification operation (high-pass channel) may not address all harmful heterophilous cases. In our ACM and ACMII framework, the node-wise channel mixing mechanism can make ACM-GNNs and ACMII-GNNs  perform at least as well as the uni-channel GNNs.
>
> **Q4**
> The filterbank based method is (1) trivial and (2) not new. (3) Adding that the theoretic analysis of diversification operation or  new metric is also (4) not convincing, (5) the novelty of this paper is incremental.
>
> **R4**
> (1) In our opinion the filterbank based method is not trivial. Could you give more detailed comments to help us to understand your view?
>
> (2)
> To our knowledge, we are the first to analyze and apply the filterbank technique in graph representation learning to deal with the heterophily problem. There are geometric scattering networks [1,2] that apply filterbanks to address over-smoothing problem. But their definition of filterbank is totally different from the one used in our paper and theirs are not designed to address heterophily problem.
>
> (3) It is not clear to us why you think our theoretic analysis of diversification operation is not convincing. If your comment is based on **Q3**, please see our response to it.
>
> (4) We have empirically and theoretically verified the advantages of the proposed metrics over the existing homophily metrics in Figure 1, Section 3.2 and Appendix B. To us the advantages over the existing homophily metrics are quite convincing. It is not clear to us why you have an opposite point of view.
>
> (5) In our paper we discuss the differences between our method and the existing methods in Section 5, explain the insights behind HP filters for the heterophily problem, generalize the definition of filterbank to get the identity channel and design a node-wise channel mixing mechanism to learn information from different channels in other sections. To our knowledge, there is no prior work that adaptively applies a 3-channel filterbank architecture to address heterophily problem. We also discussed the detailed differences of ACM and ACMII with 2 SOTA methods, FAGCN and GPRGNN in Appendix J.
>
> Our paper develops a different approach which is not a simple improvement over the existing methods. Thus, in our opinion our contribution is not incremental.
>
> [1] Gao, Feng, Guy Wolf, and Matthew Hirn. "Geometric scattering for graph data analysis." International Conference on Machine Learning. PMLR, 2019.
>
> [2] Min, Yimeng, Frederik Wenkel, and Guy Wolf. "Scattering gcn: Overcoming oversmoothness in graph convolutional networks." arXiv preprint arXiv:2003.08414 (2020).

---

### Public Comment · ~Susheel_Suresh1 · 2021-11-13
**Related work on GNNs and mixing patterns**

Hi, I wanted to bring to authors notice a related work from KDD '21 [1] which is missed in the paper and discussions. The referenced paper also analyses the behavior of GNNs w.r.t mixing patterns in networks using the notion of local assortativity. While the analysis is experimental in nature I still think it is very relevant here.

Wanted your thoughts on the experimental observations witnessed in [1] in relation to the current submission.

[1] Susheel Suresh, Vinith Budde, Jennifer Neville, Pan Li, and Jianzhu Ma. Breaking the Limit of Graph Neural Networks by Improving the Assortativity of Graphs with Local Mixing Patterns. KDD '21

Thanks

---

> ### Author Response · Authors · 2021-11-18
> **Will consider your paper**
>
> Hi Susheel,
>
> We will consider comparing with your KDD paper in the revised version.
>
> Best,
>
> Authors

---

### Public Comment · ~Benedek_Andras_Rozemberczki1 · 2021-11-14
**Chameleon and Squirrel dataset**

The datasets are not appropriate properly. This is the right paper:

@article{rozemberczki2021multi,
  title={Multi-scale attributed node embedding},
  author={Rozemberczki, Benedek and Allen, Carl and Sarkar, Rik},
  journal={Journal of Complex Networks},
  volume={9},
  number={2},
  pages={cnab014},
  year={2021},
  publisher={Oxford University Press}
}

---

> ### Author Response · Authors · 2021-11-18
> **References added**
>
> Hi Benedek,
>
> Your papers have been added to our revised version. Please take a look.
>
> Best,
>
> Authors

---

### Decision · Program_Chairs · 2022-01-20

**Decision:**

Reject

**Comment:**

The reviewers agree that the paper is addressing an interesting problem. However, the authors analyze the effect of heterophily on GNN for node classification. The authors simplify the analysis by removing the nonlinearity in the GNN model and derive some theoretical results. However, the analysis is very specific to the simplified version of GNN, and the link to later proposed solution is also weak. Furthermore, a more significant improvement in experiments will also make the paper more convincing.